# Diversity, multifaceted evolution, and facultative saprotrophism in the European *Batrachochytrium salamandrivorans* epidemic

Moira Kelly [1✉], Frank Pasmans[1], Jose F. Muñoz [2], Terrance P. Shea[2], Salvador Carranza [3], Christina A. Cuomo [2,4] & An Martel [1,4✉]

While emerging fungi threaten global biodiversity, the paucity of fungal genome assemblies impedes thoroughly characterizing epidemics and developing effective mitigation strategies. Here, we generate *de novo* genomic assemblies for six outbreaks of the emerging pathogen *Batrachochytrium salamandrivorans* (*Bsal*). We reveal the European epidemic currently damaging amphibian populations to comprise multiple, highly divergent lineages demonstrating isolate-specific adaptations and metabolic capacities. In particular, we show extensive gene family expansions and acquisitions, through a variety of evolutionary mechanisms, and an isolate-specific saprotrophic lifecycle. This finding both explains the chytrid's ability to divorce transmission from host density, producing *Bsal*'s enigmatic host population declines, and is a key consideration in developing successful mitigation measures.

[1] Wildlife Health Ghent, Department of Pathology, Bacteriology and Avian Diseases, Faculty of Veterinary Medicine, Ghent University, 9820 Merelbeke, Belgium. [2] Broad Institute of MIT and Harvard, Cambridge 02142 MA, USA. [3] Institute of Evolutionary Biology (CSIC-UPF), 08003 Barcelona, Spain. [4]These authors jointly supervised this work: Christina A. Cuomo, An Martel. ✉email: Moira.Kelly@ugent.be; An.Martel@ugent.be

Emerging fungal pathogens represent one of the greatest threats globally to human health, food security, and biodiversity. Fungal pathogens are estimated to be responsible for the loss of 15–33% of crops annually and an estimated 150 million people suffer from life-altering or terminal mycoses[1], while the frequency of novel fungal pathogen emergence is only accelerating[2].

*Batrachochytrium salamandrivorans* (*Bsal*) is one of the two chytrid fungi responsible for amphibian chytridiomycosis. Its closest relative, *Batrachochytrium dendrobatidis* (*Bd*), has been connected with the extinction or decline of over 500 species[3], representing the most severe loss of vertebrate biodiversity in recorded history. It is feared that *Bsal* may precipitate a similarly catastrophic multispecies pandemic[4]. Following its discovery in the Netherlands in 2013, *Bsal* was detected in a rapidly growing region with outbreaks identified in Belgium, Germany[5], the UK[6], and Spain[7]. Like *Bd*, *Bsal* can cause rapid mortality and possesses the rare capacity to drive host populations to functional extinction[8]. *Bsal* can also infect a broad range of amphibian hosts, and while disease is more restricted to salamander species[9], other amphibians can still act as reservoirs[8,10]. Furthermore, *Bsal* is the more common chytrid in endemic Asian urodele populations[11], possesses a resistant environmental life stage[8], and is highly prevalent in the pet trade[12] so is predicted to have a high probability of introduction into naive wild populations[13]. Together, *Bsal* clearly represents a significant threat to global biodiversity.

However, the literature around *Bsal*, bar one infection trial in one article[8], studies a single *Bsal* isolate, and to date only one poorly contiguous genome assembly has been published for *Bsal*[14]. Within fungal pathogens this is not uncommon—although recent years have seen huge progress in fungal genomics research, we still have relatively few fungal genome assemblies with a number of major fungal pathogens represented by only one or two assemblies, often 3–4 orders of magnitudes fewer than their bacterial counterparts (Supplementary Fig. 1). But there are disadvantages to assuming a single reference isolate, often a laboratory strain selected for suitability for research rather than representative of important features of the epidemic, is representative of the epidemic or species as a whole. In the case of *Bsal*, while little is known of its epidemiology and movement between populations, its relatively clustered distribution had led many to consider the epidemic a recent introduction with "all isolates currently stemming from a single epizootic clone"[15].

In this study, we sought to describe the variation present within the European outbreak, using Pacific Biosciences (PacBio) Sequel and Illumina data to assemble highly contiguous *de novo* assemblies for isolates from six outbreaks, generate a dated phylogeny, and investigate evolutionary rates and mechanisms. These per-outbreak assemblies exposed the scope of intraspecific variation in both genome size and composition, as well as isolate-specific protein acquisitions, losses, and expansions, and facilitated the investigation of *Bsal* evolution including horizontal gene transfer (HGT) events occurring after lineage divergence. These analyses guided the discovery of an isolate-specific saprotrophic lifecycle, a finding that both helps to explain *Bsal*'s devastating host population declines and necessitates a substantial redirection of current mitigation efforts, which largely focus on amphibian hosts.

## Results

**A data set representative of the European *Bsal* epidemic.** We sequenced nine *Bsal* isolates collected from six outbreak sites, all associated with fire salamander (*Salamandra salamandra*) die-offs and together spanning much of the geographical and temporal range of the European outbreak, using both PacBio and Illumina sequencing technologies (see Table 1, Fig. 1, and Supplementary Fig. 2). For three of these outbreak sites, we collected an additional strain, isolated 1–5.5 years after the initial isolation, and performed Illumina sequencing. Principal component analysis (PCA), variant, and gene presence–absence analyses of the Illumina data indicated that isolates within "time-series pairs" appear to cluster together, displaying apparently lower variation compared to that observed between outbreak sites (see Supplementary Figs. 2 and 10), and this supported considering one assembly per outbreak site as representative. PacBio long reads were assembled with multiple *de novo* assemblers (see "Methods"), with the best assembly based on contiguity and completeness selected for further analysis—namely, phased Falcon-Unzip assemblies for BundBos2013 and Luik2014 (N50 123.6–346.3 kb, BUSCO completeness scores 93.5–96.6%) and Flye v.2.4.2 assemblies for Captive2015-1, Captive2015-2, Rob2015, and Catalan2018 (N50 74.9–203.2 kb, BUSCO scores 91.0–97.6%).

**Multiple clades, with outbreak clusters.** We inferred a RAxML phylogeny using Illumina data (Fig. 1) and the *Bd* Jel423 isolate as an outgroup and found that the Spanish Catalonian isolate is more closely related to the Captive2015 isolates, with the isolates from the Bunderbos and Robertville outbreaks forming a second cluster and the Luik outbreaks forming a third cluster. Identity by descent analyses (Supplementary Fig. 3B), and single-nucleotide polymorphism (SNP) PCA analysis (Supplementary Fig. 2) agreed with this phylogeny. However, we also inferred a Beast2 SNAPP phylogeny (Supplementary Fig. 3C), without *Bd* Jel423 as an outgroup, which shows the Luik2014 isolate as the first to branch from the phylogeny and dated the root of these clades as diverging 58,416 years ago (95% highest posterior density (HPD) range 41–4.6 × 10$^8$ years). Phylogenetic analysis inferred using core ortholog sequence by OrthoFinder[16] based on protein annotations from PacBio assemblies also place the Luik2014 isolate as an early diverging clade with the BundBos2013 isolate next to diverge; however, this protein sequence-based approach positions the Rob2015 isolate in a clade with the Catalan2018 and Captive2015 isolates (see Supplementary Fig. 3A).

**Table 1 Summary of isolates.**

| Official naming scheme | Alias | Year of isolation | Location |
|---|---|---|---|
| AMFP13/1 | BundBos2013 | 2013 | Bunderbos, Netherlands |
| AMFP14/1 | Rob2014 | 2014 | nr. Robertville, Belgium |
| AMFP14/2 | Luik2014 | 2014 | nr. Liege, Belgium |
| AMFP15/1 | Captive2015-1 | 2015 | Captive |
| AMFP15/2 | Captive2015-2 | 2015 | Captive |
| AMFP15/3 | Rob2015 | 2015 | nr. Robertville, Belgium |
| AMFP17/1 | Luik2017 | 2017 | nr. Liege, Belgium |
| AMFP18/1 | BundBos2018 | 2018 | Bunderbos, Netherlands |
| AMFP18/2 | Catalan2018 | 2018 | Montnegre i el Corredor Natural Park, Catalonia, Spain |

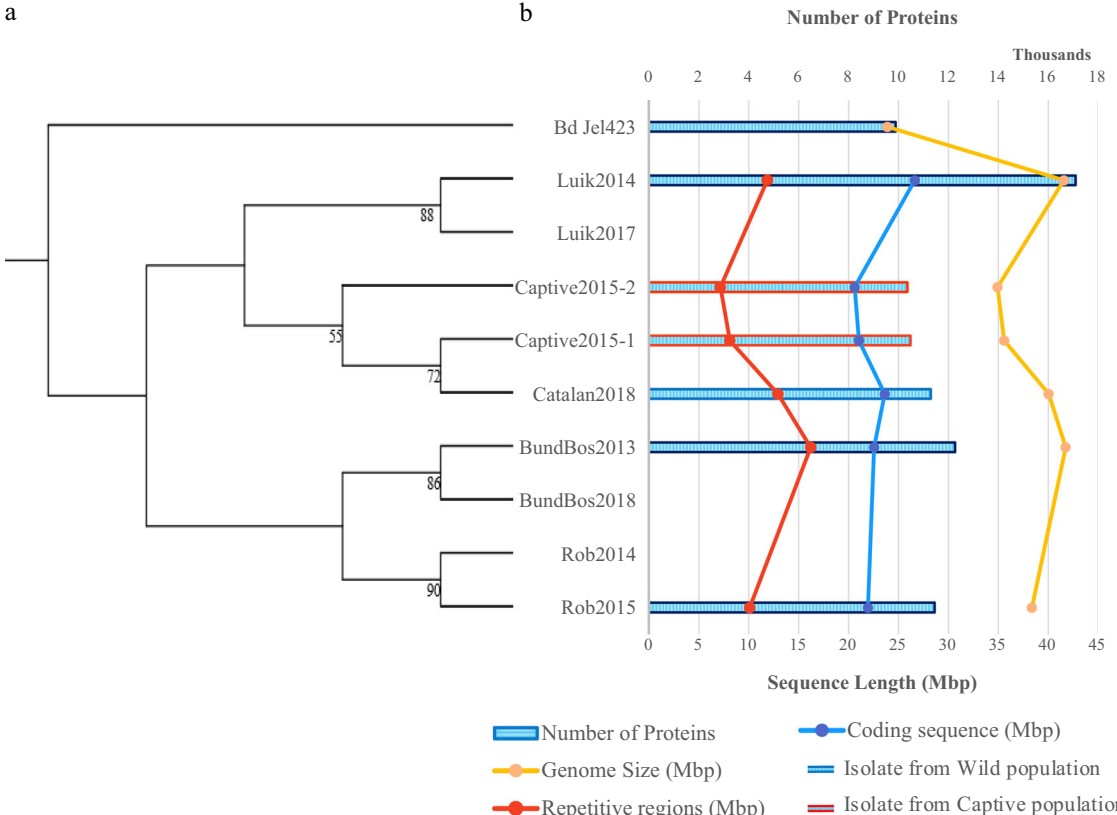

**Fig. 1 Summary of genomic features, location and phylogeny of *Batrachochytrium salamandrivorans* isolates. a** RAxML inferred phylogeny constructed using 36,495 SNPs in linkage equilibrium, including *Bd* Jel423 isolate (PRJNA13653, GCA_000149865.1, Farrer et al.[14]) as an outgroup; values represent bootstrap support; **b** isolate assembly summary where bars represent gene count and lines represent total repetitive sequence length (red), total coding sequence length (blue) and genome size (yellow). BundBos2013 represents the ground zero isolate from the Netherlands in 2013; BundBos2018 was isolated from the same outbreak population in 2018; Luik2014 is from an outbreak near Liege, Belgium in 2014; Luik2017 was isolated from the same outbreak in 2017. Rob2014 is from an outbreak in 2015 near Robertville, Belgium, <40 km from Liege, and Rob2015 was isolated from the same outbreak in 2015. Catalan 2018 was isolated from an outbreak in the Montnegre i el Corredor Natural Park, Catalonia, Spain. Captive population locations cannot be disclosed in order to maintain anonymity, both were isolated in 2015.

***Bsal* genomes are highly variable**. Our data improve upon the single previously published[14] *Bsal* genome (N50 of 346.3 vs 10.5 kb, see SI Table 1) and found the BundBos2013 isolate to be 9 Mbp larger than previously reported with 41.6 Mbp of primary contigs in our phased diploid Falcon-Unzip assembly. We found considerable variation in genome size (range 34.9–41.6 Mbp), repetitive sequence content (7.1–16.2 Mbp), and gene count (10,353–17,091) within the European *Bsal* epidemic (Fig. 1). Within the Chytridiomycetes, our *Bsal* genomes place among the larger assemblies with some of the highest gene counts in relation to assembly size (see Supplementary Fig. 4).

**Diverse gene arsenals**. The *Bsal* assemblies exhibit very variable gene complements due to clade- and isolate-specific acquisitions, losses, and expansions. The number of MEROPS protease[17] homologues varies nearly threefold between isolate assemblies (SI Table 2), with a greater than twofold variation in copy number of some peptidase families, including the M36 metalloproteases (range 89–202 copies), which are commonly fungal pathogen effector proteins, and the A28A subfamily (range 5–79 copies), which is known to regulate the cell cycle and protein secretion (see Fig. 2, Supplementary Table 2, and Supplementary Data 1). Isolate assemblies show similarly varied counts of Carbohydrate Binding Molecules (CBMs) CAZyme proteins, particularly those involved in chitin metabolism and remodelling (see Fig. 2a,

Supplementary Fig. 5, and Supplementary Data 1). Clustering *Bsal* proteins by homology using OrthoFinder[16] indicated large copy number expansions as well as hundreds of protein clusters unique to single isolates (range 15–909 per isolate, see SI Table 3). Maximum likelihood gene trees of select orthogroups and candidate effector proteins revealed various expansions occurred before, during, or after isolates diverged (see Fig. 2c, d and Supplementary Fig. 6). However, even within this context of extensive intraspecific variation, the Luik2014 isolate is particularly conspicuous, containing genes from 20 MEROPS protease families absent in the other isolates (see Fig. 2b and Supplementary Data 1). Ortholog clustering indicated 909 orthogroups unique to the Luik2014 isolate, containing hundreds of candidate effector proteins without homologues recognised in other *Bsal* assemblies or *Bd* Jel423 (SI Table 3). Such extensive gene family expansion and divergence is resource expensive and so may be associated with functional divergence, suggesting potential isolate-specific capacities[18,19].

**Isolate-specific saprotrophic capacity**. Examining the gene family expansions, one striking difference between the isolates were differential copy numbers of 17 protein families associated with plant carbohydrate metabolism or plant pathogenicity (see Fig. 3a). In particular, the Luik2014 isolate has, relative to other sequenced Chytridiomycota species, either acquired or uniquely retained (SI

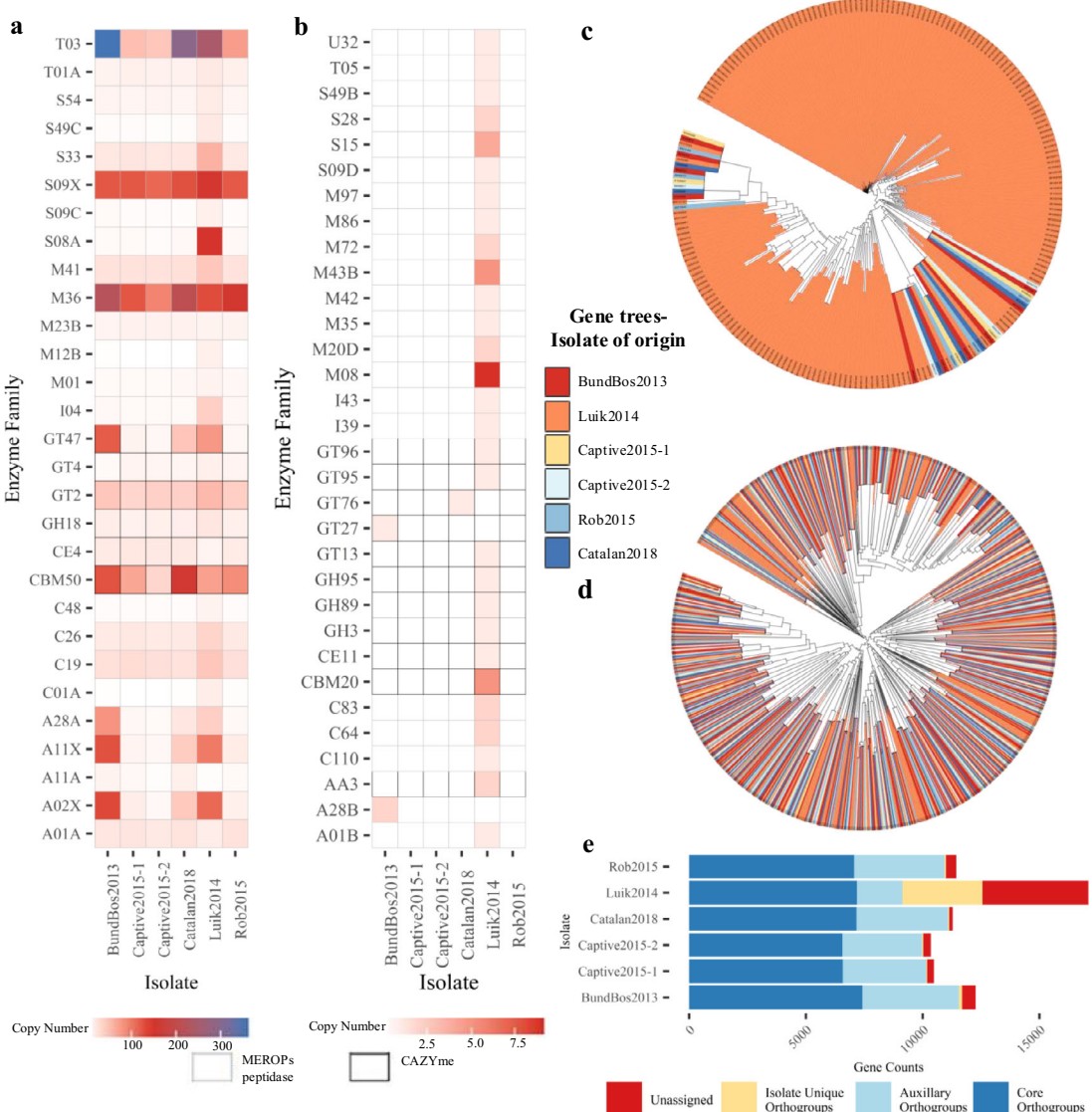

**Fig. 2 Variation in isolate protein annotations.** Heatmap showing **a** the number of copies of predicted CAZymes and MEROPS proteases with highly variable copy numbers; **b** the number of copies of MEROPS peptidase or CAZyme families unique to an isolate; **c** FastTree inferred gene tree of candidate serine endopeptidases in the S8A MEROPS protease family, a family of secreted endopeptidases associated with nutrition, showing expansion in the Luik2014 isolate following isolate divergence (148 copies in Luik2014, indicated here with orange coloured clades and labels, compared to 4–6 copies in all other isolates, coloured according to the gene trees legend); **d** the FastTree gene tree of candidate Crinkler proteins, which by comparison have undergone large expansion and divergence within *Bsal*, the majority of which occurred before isolate divergence resulting in most clades containing sequence from numerous or all isolates, also coloured according to the gene trees legend. Clustering Crinkler candidates using CDhit identified 142 clusters with 2+ proteins, with the largest cluster containing only 9 proteins from multiple isolates indicating limited isolate-specific expansion; **e** bar chart showing the number of orthogroups identified by OrthoFinder that are: core—at least one copy present in all isolates, auxiliary—present in >1 isolate, unique—only present in that isolate, and unassigned—genes unassigned to a orthogroup.

Tables 4 and 5) proteins from nine CAZyme families, four of which (GH3, GH95 and CBM20 family proteins, and AA3 cellobiose dehydrogenase) are implicated in cellulose, xylan, pectin, and starch metabolism during fungal saprotrophism and plant pathogenesis[20–22]. Considering these protein acquisitions and expansions together with the ancestral saprotrophic role of chytrid fungi, we hypothesised that some *Bsal* isolates might be able to nutritionally utilise plant material. We tested this capacity using three model systems. First, we observed growth in a lima bean (*Phaseolus lunatus*) culture medium (LB), comparing three measures of *Bsal* growth: motile spore count, sporangia count, and sporangia coverage. We found the Luik2014 isolate actually grew better in LB compared to standard tryptone–gelatin

hydrolysate–lactose (TGhL)[9] medium (Luik2014 sporangia count: TGhL medium incidence rate ratio (IRR) = 0.26, confidence interval (CI) = 0.14–0.48, Wald Test $p = 1.61e{-}05$; Luik2014 sporangia Coverage proportion: TGhL: time IRR = 0.82, CI = 0.70–0.97, Wald Test $p = 0.023$). All other isolates exhibited better growth in TGhL (TGhL sporangia count IRR = 1.79, CI = 1.16–2.76, Wald Test $p = 0.009$, TGhL sporangia coverage over time IRR = 1.31, CI = 1.17–1.48, Wald Test $p = 7.38e{-}06$) (see Fig. 3b, Supplementary Fig. 7, and SI Table 6). Spore count showed no effect with isolate identity, medium, or time. Second, we tested growth on autoclaved entire plant material to study the utilisation of plant nutrients in a more complex form. The Luik2014 isolate was the only *Bsal* strain to consistently grow on the

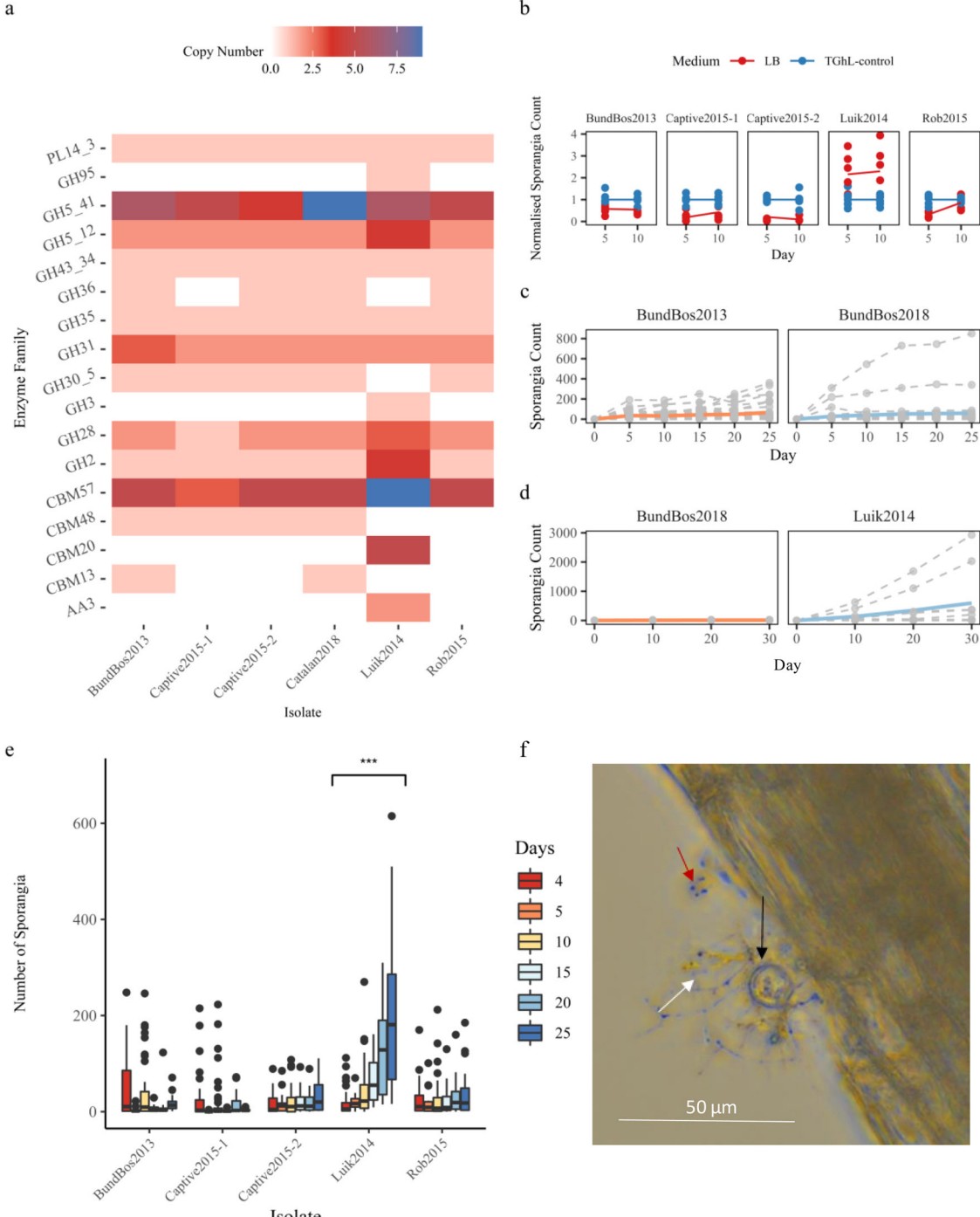

**Fig. 3 Intraspecific differential vegetative metabolic capacities. a** Heatmap displaying copy numbers of enzyme families associated with metabolising plant carbohydrates; **b** scatterplot of sporangia counts in cultures grown in lima bean (LB) medium (red) vs optimised conditions of TGhL medium (blue), normalised for each isolate and time against the average count for the optimised TGhL medium, with the average for each condition, isolate and time represented by a solid line; **c** Line graph of counts of mature sporangia visualised attached on each sample of plant material at days 5-25 for BundBos2013 and BundBos2018, solid line depicts average **d** growth plots of Luik2014 and BundBos2018 on autoclaved beech leaf material, dashed lines represent each individual sample, solid line represents averaged count. **e** Box plot representing counts of mature sporangia visualised attached on sterilised hay plant material at days 4–25 per isolate. Boxplots constructed with the centre representing the median, the bounds of the box representing 25th and 75th percentiles, whiskers representing the biggest or smallest value within 1.5× interquartile range of 25th/75th percentiles and data points outside this as outlier points. Significant increase in sporangia count over time, compared to other isolates, indicated by three asterisks as $p < 0.001$ (Wald Test $p = 5.73e-13$, $n = 230$ (46 biologically independent replicates per isolate across 4 experiments)). **f** Light microscopy of BundBos2018 sporangia attached to autoclave hay material (20× magnification), photograph taken 327 days after introduction of spores to material, black arrow indicates sporangia, white arrow indicates rhizoids that are extensive compared to in culture, red arrow indicates small new sporangia in which moving spores could be visualised.

"complex" autoclaved hay plant material model (Luik2014 Count*Time IRR = 1.18, CI = 1.13–1.23, Wald Test $p$ = 5.73e−13, $n$ = 230 (46 replicates per isolate); see Fig. 3e and Supplementary Fig. 8). The two other wild isolates (BundBos2013 and Rob2015) sporadically exhibited growth with, for example, 4 out of 46 BundBos2013 samples exhibiting growing mature sporangia counts in autoclaved hay experiments. However, these four BundBos2013 samples did continue to produce motile-spore-filled mature sporangia on autoclaved hay for >10 months, indicating that this isolate has the potential for sustained growth on plant material. By contrast, the two isolates from captive populations (Captive2015-2 and Captive2015-1) showed no capacity for sustained growth in any experiment (nearly 150 samples per isolate in total).

As we hypothesised that this phenotype could sustain *Bsal* populations following host population collapse, we compared the saprotrophic capacity of *Bsal* isolated 5.5 years after host population collapse (BundBos2018) with that of the *Bsal* isolate from the start of the outbreak (BundBos2013). We found that, although fewer BundBos2018 spores initially attached (Bund-Bos2018 count IRR = 0.57, CI = 0.37–0.87, Wald Test $p$ value = 0.009), there was a weak positive effect on the sporangia count over time (BundBos2018:time IRR = 1.03, CI = 1.003–1.06, Wald Test $p$ = 0.027, see Fig. 3c, f and Supplementary Fig. 9A). Finally, we wanted to test this capacity in a more ecologically relevant setting, so we tested the growth of the two isolates with the greatest growth capacity (Luik2014 and BundBos2018) on autoclaved beech leaves (*Fagus sylvatica*), a vegetative material abundant in the Northern European outbreak sites. Both isolates grew on this material, with the Luik2014 isolate showing notably higher initial counts when compared to hay samples (average 135.9 vs 41.5 at day 10) and compared to the BundBos2018 isolate (BundBos2018 count IRR = 0.13, CI = 0.04–0.45, Wald Test $p$ value = 0.001, $n$ = 20, see Fig. 3d). However, we saw no difference in growth rate between isolates (count over time IRR = 1.04, CI = 1.02–1.06, Wald Test $p$ = 0.0002, $n$ = 20, see Supplementary Fig. 9B). Thus, *Bsal* was able to attach to and grow on environmentally relevant plant materials. We observed notable variation in saprotrophic capacity both between- and within-isolate treatment groups, suggesting substantial potential for adapting to variable environmental conditions, such as amphibian host population collapse.

**A plethora of mechanisms for rapid evolution**. To better understand how *Bsal* evolves, we studied mechanisms of divergence within and between outbreak sites. Comparing two isolates (BundBos2013 and BundBos2018) collected 5.5 years apart from the index site we observed a substitution rate of $8.25 \times 10^{-5}$ per site per year, which is faster than many fungal mutation rates listed in the literature[23–26]. We identified 6863 insertion/deletion (indel) events, affecting 959 genes between the 2013 and 2018 isolates, which are notably higher than the 470 indels in 149 genes observed over a comparable time frame in *Bd*[23]. In isolate pairs collected from two other outbreak sites 1 year (Rob2014 and Rob2015) and 3 years (Luik2014 and Luik2017) apart, we see even faster evolutionary rates of $3.97 \times 10^{-4}$ mutations per site per year and 6688 indels affecting 814 genes (Rob isolates), and $3.49 \times 10^{-4}$ mutations per site per year and 8194 indels affecting 6238 genes (Luik isolates). However, as this timescale is unlikely to allow for the effective removal of deleterious mutations by selective pressure, and the source population diversity remains unknown, with some of this divergence potentially occurring before sampling within these lineages, these estimates likely represent an overestimate of the long-term mutation rate[27];

indeed our Beast2 SNAPP phylogeny converged on a clock rate of $5.4 \times 10^{-6}$ (95% HPD $8.29 \times 10^{-14}$–$1.63 \times 10^{-4}$).

Read depth, normalised by isolate and GC content, indicates that 33 out of 233 primary contigs of the BundBos2013 assembly displayed copy number variation (CNV) between BundBos2013 and BundBos2018. We found 18 contigs (6.35 Mbp, 2012 genes) decreasing in copy number in BundBos2018 and 15 contigs (8.43 Mbp, 2570 genes) increasing in copy number in Bund-Bos2018. This large-scale variation in copy number suggests that chromosomal aneuploidy could be an underlying mechanism. We used two-tailed Fisher exact tests for enrichment to look for protein family domains (Pfam) that were disproportionately represented on contigs with varying copy number, as a mechanism for altered expression. We found 14 Pfam domains enriched ($q < 0.05$) in contigs with lower copy number in BundBos2018 and 17 Pfam domains enriched in contigs with increased copy number in BundBos2018 (see Supplementary Data 2). A number of the plant biomass degradation enzymes displayed CNV within our time-series isolates with 25 of the 58 genes present in the BundBos2013 assembly located on 13 contigs displaying CNV—17 genes on contigs with increased copy number in BundBos2018 and 8 on contigs with decreased copy number.

To see whether gene presence or absence data suggested greater stability in gene arsenal within an outbreak than between outbreaks, we identified candidate regions of deletion or duplication using CNVnator[28]. To allow comparisons between isolates from the same outbreak site vs isolates from different outbreak sites, we ran CNVnator on aligned bam files generated by aligning Illumina reads from all isolates to the BundBos2013 assembly, as well as aligning the Luik2014 and Luik2017 isolates to the Luik2014 assembly, and the Rob2014 and Rob2015 isolates to the Rob2015 assembly. We then performed pairwise comparisons of the size, and number of affected genes, of regions identified as deleted or duplicated in one of the isolate pairs (see "Methods" for more details and Supplementary Fig. 10). We found that the length of differentially deleted genome sequence in pairwise comparisons was smaller in isolate pairs from the same outbreak site than between outbreak sites ($T$ test $t = -2.873$, df = 33.9, $p = 0.0069$, on average 312 kbp differentially deleted in pairs from the same outbreak site vs an average of 487 kbp differentially deleted in pairs from different outbreak sites). The number of genes deleted were not significantly higher or lower between pairs from the same outbreak site compared to pairs from different outbreak sites (T-test $t = -0.93128$, df = 17.2, $p = 0.36$, on average 198 genes differentially deleted in pairs from the same outbreak site vs on average 245 genes differentially deleted in pairs from different outbreak sites). However, both the size of differentially duplicated sequence and the number of differentially duplicated genes were found to be smaller in isolate pairs from the same outbreak site compared to pairs containing isolates from different outbreak sites ($T$ test duplicated region sequence size $t = -3.2644$, df = 25.91, $p$ value = 0.003079, gene count $t = -2.3485$, df = 19.961, $p$ value = 0.02926; mean 607 kbp and 282 genes differentially duplicated in pairs from the same outbreak site, mean 1.54 Mbp and 578 genes differentially duplicated in pairs different outbreak sites). Thus, while we see substantial variation within an outbreak site, it appears that there is more diversification in the gene arsenal between isolates from different outbreak sites than isolates from the same outbreak site. However, the limited sample size studied here, with only pairs of isolates sequentially obtained from each outbreak site, does limit such comparisons, and it is possible that the apparent greater diversity observed between vs within outbreaks is an artefact of this limited sampling.

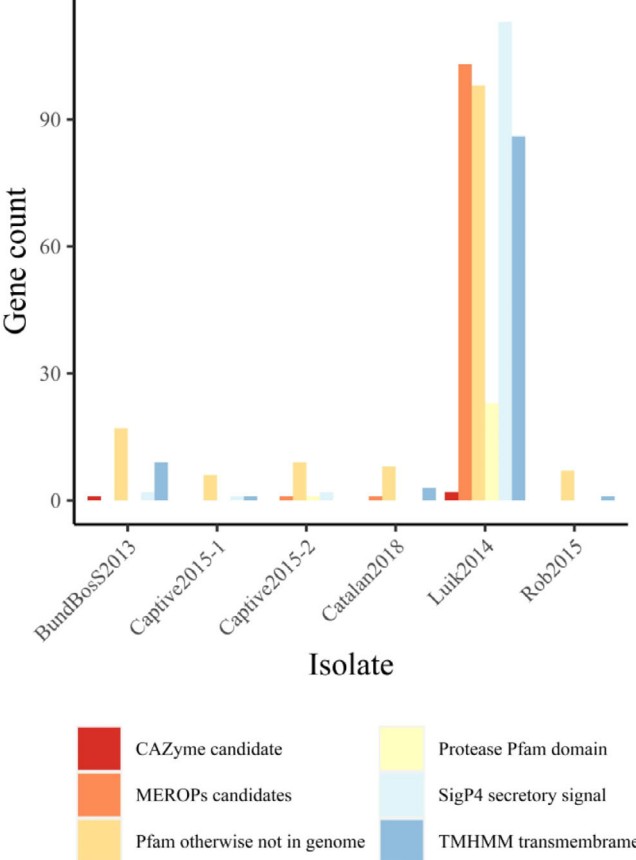

**Fig. 4 Variation in genes identified as candidates for horizontal gene transfer.** Comparison of number of horizontal gene transfer candidate genes, determined using phylogenetic techniques, and their annotations. Given that not all HGT candidates were annotated within one of these groups, and some maybe be included in more than one of these categories (e.g. a protease Pfam and a Pfam otherwise not present within the assembly), thus the total HGT candidate counts do not equal the sum of the depicted columns: BundBos2013 (total 43 candidates), Luik2014 (509), Captive2015-1 (14), Captive2015-2 (16), Catalan2018 (16), Rob2015 (14).

On a greater timescale, we observed that isolates from different outbreaks have undergone different mechanisms of genome modification and expansion. For example, with the two larger assemblies BundBos2013 and Luik2014, while BundBos2013 has expanded repetitive regions (16.2 Mbp of repetitive regions vs 11.9 Mbp in the Luik2014 assembly and 7–10 Mbp in the other isolates—SI Table 7), the Luik2014 isolate shows expanded protein-coding regions (26.6 Mbp of sequence coding for 17,091 proteins, vs 22.6 Mbp of sequence coding for 12,269 proteins in the BundBos2013 isolate, SI Table 8). There is evidence of transposable element (TE) expansion in both the BundBos2013 and Luik2014 isolate, with 4.8 Mbp and 5.9 Mb, respectively, annotated as long terminal repeats and long interspersed nuclear elements (compared to e.g. 732 kbp in the Captive2015-2 isolate) and gene annotations indicating expansions in protease families associated with TEs with, for example, 253 BundBos2013 and 206 Luik2014 genes annotated as MEROPS A02X or A11X peptidase families, compared to 13–78 genes in the other *Bsal* isolate assemblies. However, we found little evidence of effector protein duplications and expansions linked with islands of low GC and highly repetitive content as hypothesised by the "two speed" genome (for further discussion, see Supplementary Information). Phylogenetic methods assessing species relationships indicated 509 horizontally transferred candidate proteins in the Luik2014

isolate compared to 43 in the BundBos2013 assembly and 14–16 in the other assemblies. These candidates include a number of novel Pfam domains and candidate effector proteins as well as, for example, two Acetyl xylan esterases, a vegetative biomass degradation enzyme, in the Luik2014 isolate, which shares highest sequence similarity with proteins from Gram-negative deltaproteobacteria or acidobacteria species (see Fig. 4, Supplementary Fig. 11, and Supplementary Data 3). A small number of these candidates are present in all or most isolates, some are clade-specific, but most appear to have occurred since Luik2014 diverged from the other isolates. Thus, a variety of mechanisms have contributed to the acquisition of divergent genomic content within the European *Bsal* epidemic.

## Discussion

Through a combination of phylogenetic, genomic, and phenotypic methods, we identified vast variation within the European *Bsal* epidemic. These results are unexpected based on epidemiological characteristics, and particularly its limited geographical and temporal range[15]. Our Bayesian molecular dating converged on >50,000 years ago for the major clade divergence; however, additional samples are needed to decrease the confidence intervals for dating estimates. Dating using mitochondrial sequence (as performed for *Bd*[29]) was attempted but unsuccessful (see "Methods" for details). Additionally, we are more likely to have underestimated the time since isolate divergence given that our dating input are based on recent mutation rates that are often overestimated due to inadequate time for the selective removal of deleterious mutations[27], and as SNAPP stipulates a strict molecular clock this may be complicated should branches experience different mutation rates e.g. before and after introduction.

It seems unlikely that the observed diversity represents purely historical divergence and a single recent introduction of a diverse population into Europe (so sampling resultant diversity present within populations at all sites) as isolates collected serially from the same outbreak site consistently cluster together, appearing to show less divergence within- than between-outbreak sites (see SI Fig. 2 and Supplementary Fig. 3A). While further sampling from within outbreaks would be needed for more definitive conclusions; unfortunately the speed with which *Bsal* decimates populations[8], with minimal to no recovery years after introduction, heavily impedes sequential sampling. However, these results do suggest that *Bsal* may have been present in Europe for considerably longer than previously thought, that this epidemic represents multiple introductions, or indeed both. The prevalence of *Bsal* within the pet trade[10,12], together with the pervasive presence of introduced invasive species in European amphibian assemblages[30–32], suggest that exotic, potentially infected animals have frequently entered European amphibian communities. Indeed the recent *Bsal* outbreak in Catalonia, Spain is considered to be associated with the introduction of captive invasive species[7], and this isolate nests with isolates from captive amphibian populations in our phylogenetic analyses. Furthermore, recent publications revealed that introductions into Europe likely occurred before the official "index" outbreak[33], and that *Bsal* outbreaks can be associated with cryptic population losses without notable mass mortality events[34]. Thus, other introductions and subsequent population collapses may have occurred undetected.

Using long read technology, we significantly improved upon the single previously published[14] *Bsal* genome (see SI Table 1), with our BundBos2013 assembly displaying improved contiguity (N50 of 346.3 kb vs 10.5 kb), a concomitant increase in gene counts (12,269 vs 10,138) and improved resolution of repetitive elements (16.2 vs 5.9 Mbp). While our assemblies are still

fragmented, the variations in genome size and content observed between isolates are substantial and consistent with different assemblers (Supplementary Fig. 12). We see isolate-specific expansions in many genes known to be required for pathogenesis in other fungal pathogen–host systems such as Crinkler proteins[35], M36 metalloproteases[36,37], and genes involved in chitin metabolism[38]. Isolate-specific protein family expansions, particularly in effector proteins, may suggest different pathogenic strategies and competencies[18,19].

As well as expansions in known effector proteins, our assemblies indicated expansions in numerous metabolic functions, particularly in the Luik2014 isolate. This guided the hypothesis that isolates may have differential metabolic capabilities, particularly of vegetative material, and led to the discovery of saprotrophic potential in some isolates. Notably, isolates for which vegetative growth has been observed have also all shown fatal pathogenicity in both their origin amphibian populations and sentinel animals under laboratory conditions such that the vegetative lifecycle appears to be in addition to, rather than instead of, the pathogenic state. Although saprotrophic capacity complements the natural history of the chytrid phylum[39], it was unknown in the amphibian chytridiomycosis-causing Batrachochytrium species and has significant ecological implications. First, it is likely a key facilitator in continuing the drive to extinction after host population collapse[40]. Second, as this capacity is isolate-specific, it suggests that not every introduction may represent the same risk to amphibian populations. Finally, this knowledge is key for instructing conservation efforts as the growth of Bsal on ubiquitous dead plant material implies eliminating the pathogen cannot rely upon removing amphibian populations, as has been applied in recent outbreaks[7]. Future outbreak responses should be based upon outbreak-specific data and, in the case of saprotrophically capable Bsal isolates, elimination and mitigation efforts must also address potential environmental reservoirs (for further discussion, see Supplementary Information). Given the significant challenges, substantial expense, and likely collateral ecosystem damage this represents, together with currently limited evidence for effective environmental spread[41], this highlights the urgent need to prevent further introductions.

As captive amphibian collections are thought to be key for vectoring Bsal internationally and into naive environments[10,12], it is interesting that isolates from captive amphibian populations possess notably fewer isolate "unique" clusters and proteins predicted to be effector proteins (see SI Table 2) and appeared to lack the vegetative capacity. It may be that effector proteins and the vegetative phenotype is lost within the host-abundant environment of captivity, or that following introduction, only saprotrophically capable isolates with more effector proteins can become established. Alternatively, such effector protein candidates and phenotypes may be acquired following introduction, perhaps in response to the more varied host interactions and environmental conditions experienced during a wild outbreak—particularly during or following host population collapse.

Furthermore, we found that Bsal can, through a variety of mechanisms, quickly evolve and may frequently acquire genes with novel functions. For example, the isolate most competent at growing on vegetative material had acquired genes predicted to be vegetative biomass degrading enzymes through HGT. Within an outbreak, we observed CNV in several genes associated with the metabolism of plant material and a concurrent increased competency at nutritionally utilising plant materials. Thus, samples should be repeatedly re-isolated from established epidemics to accurately assess the variation present within outbreaks and scrutinise how this changes over time.

The emergence of novel infectious diseases from neglected phylogenetic clades presents numerous challenges, particularly in the scope of unknown characteristics. Genomic studies already make vital contributions to tackling such pathogens[42], and the application of per-outbreak de novo genome assembling can further this by improving intraspecific comparisons without imposing the assumption that a reference isolate represents the species or epidemic as a whole. Applying this to the European Bsal epidemic, we identified vast intraspecific variation, with isolate-specific expansions, acquisitions, and losses of genes associated with diverse functions, including key disparities between Bsal strains isolated from wild and captive amphibian populations. Notably, even within a temporally and geographically restricted epidemic we found isolate-specific lifecycles and environmental niches. This knowledge is both crucial for developing successful mitigation plans and helps to explain why, contrary to most vertebrate pathogens, Bsal population dynamics can divorce from host population density and drive amphibian populations to extinction.

## Methods

**DNA extraction and sequencing.** Genomic DNA was extracted from 9 Bsal isolates (BundBos2013, Rob2014, Luik2014, Captive2015-1, Captive2015-2, Rob2015, Luik2017, BundBos2018, and Catalan2018, locations of wild populations shown in Supplementary Fig. 2; locations of captive isolates Captive2015-1 and Captive2015-2 are not included to maintain anonymity). All isolates were collected from infected fire salamanders (S salamandra) or a marbled newt (Triturus marmoratus) for the Catalan2018 isolate, naturally infected during their respective outbreak. Isolates were collected using protocols as described in Martel et al.[9]; 1–2 mm skin sections are dragged through agar medium to remove surface contamination, before being placed in TGhL medium with 200 mg/l penicillin-G and 400 mg/l streptomycin sulfate antibiotics and incubated at 15 °C. DNA was extracted from cultures of comparable passage number; all isolates were cryopreserved, revived, and cultured under identical conditions. This study complies with all relevant ethical regulations for animal testing and research. No animal experiments were performed for this paper; references to animal mortality in infection trials refer to studies previously published[4,9,43] and approved by the ethical committee of the Faculty of Veterinary Medicine (Ghent University, EC2013/10; EC2013/79; EC2015/86). Isolations were performed on naturally infected sick animals that had to be euthanised following welfare guidance.

We used the yeast protocol for Qiagen 100G genomic tips and dsDNA concentration was checked using the Qubit dsDNA HS Assay Kit. Six isolates were sequenced using PacBio Sequel technology (BundBos2013, Luik2014, Captive2015-1, Rob2015, Captive2015-2, and Catalan2018; see SI Table 9 for read statistics); for these isolates, library prep, quality checks, and sequencing on a PacBio Sequel machine were performed by BaseClear (BaseClear B.V., Sylviusweg 74, 2333 BE LEIDEN, The Netherlands, samples BundBos2013, Luik2014, Captive2015-1, Rob2015, Captive2015-2) or Novogene UK (25 Cambridge Science Park, Milton Road, Cambridgeshire, CB4 0FW, UK, sample Catalan2018). For two of the isolates (Luik2014 and BundBos2013), quality checks following library prep indicated the presence of shorter fragments and so size selection was performed on reads from these samples before sequencing.

All the above isolates, plus three isolates representing "time-series pairs" (BundBos2018—isolated 5.5 years after BundBos2013, Rob2014—isolated 1 year before Rob2015, and Luik2017—isolated 3 years after Luik2014) were also sequenced with Illumina sequencing technology. For BundBos2013, BundBos2018, Rob2014, and Rob2015, NxtGnt (NXTGNT Ghent University, Faculty of Pharmaceutical Sciences, Ottergemsesteenweg 460, B-9000 Gent, Belgium) performed the library preparation using the NEBNext Ultra II Kit, using dual indexes and six PCR cycles followed by sequencing on one lane of a Hiseq PE150 Illumina machine. For all other isolates, this was performed by Novogene UK (25 Cambridge Science Park Milton Road, Cambridgeshire, CB4 0FW, UK), with library prep also performed using the NEBNext Ultra II Kit; although with no PCR cycles for these samples, sequencing was performed on the Novaseq6000 PE150 Illumina machine. Data were visually assessed using FastQC[44].

**Genome assembly.** Isolates sequenced using PacBio Sequel technology before 2018 (BundBos2013, Luik2014, Captive2015-1, Captive2015-2, Rob2015) were assembled using both HGAP v4 from SMRT Link v5.0 using default parameters and an input genome size of 32 Mbp (estimated from Farrer et al.[14]) and Falcon Unzip[45] of Falcon 0.5 (configuration parameters optimised as per the Falcon 0.5 documentation[46] to a length cut-off of 50 bp, genome size of 37 Mbp, read length cut-off for overlapping the preassembled reads of 2500). Both these assemblies were then polished using Quiver. One isolate (Rob2015) was also assembled using the HGAP v4 followed by Falcon Unzip; however, analysis showed this assembly to be very similar to that produced by the Falcon Unzip-Quiver pipeline and so the HGAP v4-Falcon Unzip process was not repeated for the other isolates. Overall assembly quality was assessed using the GAEMR genome analysis package (http://www.broadinstitute.org/software/gaemr/)

and BUSCO scores using Busco v3 with the fungi_odb9 database (see SI Tables 1 and 8) and comparisons made to previously published BundBos2013 assembly (GCA_002006685.1). Following annotation, it became clear that three of the Falcon assemblies (Captive2015-2, Captive2015-1, Rob2015) had considerably lower BUSCO scores than the BundBos2013 and Luik2014 assemblies (67.2, 71.7 and 63.4 vs 96.6 and 93.4, respectively; see SI Table 8), suggesting that these assemblies were less complete. These samples and Catalan2018 were then assembled using Flye v2.4.2 release[47] with an estimated genome size input of 35 Mbp. Assessments indicated the Flye assemblies to be more complete than the Falcon Unzip assemblies, while the Falcon-Unzip assemblies to be the best for BundBos2013 and Luik2014, and so these assemblies were included in future analyses. The inclusion of mitochondrial sequence in the assembly was checked using GAEMR and by performing a Blastn against the *Bd* mitochondrial sequence (NCBI Accession DS022322). Further details assessing genome phasing and variant and repetitive element content can be found in Supplementary Information.

A candidate mitochondrial genome was assembled by first constructing an assembly from the PacBio Sequel reads for the Rob2015 sample using Canu[48] v1.5 (input genome size 45 Mbp). Candidate contigs were identified by the presence of mitochondrial genes from a Blastn analysis using BLAST[49] v2.2.30 with an e-value of 1e−5 against the gene annotation for the *Bd* Jel423 Mitochondrial contig (DS022322.1), together with a divergent GC content and higher coverage than the genomic sequence. We attempted to assemble a more contiguous mitochondrial genome using only reads aligning to these candidate contigs and Canu using variable genome size inputs but could not improve upon the contiguity. As such, the assembled mitochondrial genome comprises 6 contigs amounting to 589 kbp containing all "required" mitochondrial genes[50] (see Supplementary Figs. 13 and 14), which appears highly extended for a fungal mitochondrial genome. We analysed the contigs using RepeatMasker v4.0.5[51] with the RepBase v24.1 fngrep.ref library and found only 6.6% of the contigs represented repetitive elements, comprising entirely simple repeats (3.08%) or low complexity repeats (3.54%).

**Variant calling.** Aligned bam files generated by BWA v0.7.12 mem (Illumina data) and bwasw (PacBio data) and processed using Picard tools v1.792 (http://picard.sourceforge.net/) AddOrReplace, MarkDuplicates, SortSam, CreateSequenceDictionary and ReorderSam. For PacBio alignments, variants were called on Falcon Unzip assemblies using the SMRT Tools VariantCaller from SMRT Link v5.1.0 and mpileup of Samtools v1.9 and Bcftools[52] v1.8 call function. Only variants detected by both Samtools mpileup and VariantCaller were included in further analysis as a higher confidence subset. Variants were called on Illumina data sets using Samtools mpileup and the GATK v3.7.93 HaplotypeCaller, the output GVCFs were combined using CombineGVCFs, genotyped using GenotypeGVCFs, separated into SNP and Indel variants for filtering using SelectVariants, and filtered using VariantFiltration with filters "QD < 2.0 ||FS > 60.0||MQ < 40.0|| MQRankSum <−12.5||ReadPosRankSum <−8.0" for SNPs and "QD < 2.0 ||FS > 200.0||ReadPosRankSum <−20.0" for Indels. The output vcf files from both PacBio and Illumina data were filtered using VCFtools[53] v0.1.10 to remove variants with <20 read depth and for some analyses to remove non-biallelic sites. Variants were annotated using SNPeff[54] v4.1 with default parameters.

The degree of heterozygosity was estimated by categorising those for which the percentage of reads aligning to an allele was between 20 and 80% as heterogenous and defining homozygous regions as those for which <20% or >80% reads aligned to a single allele.

**Comparing time-series isolates.** Contig copy number variation between BundBos2013–BundBos2018 and Rob2014–Rob2015 isolate pairs was identified by first calculating read depth, normalised by GC and isolate average read depth per 500 bp windows (following a trial with various window sizes). We then performed two-tailed Student's t tests per contig to identify significant increases or decreases in read depth, with multiple test correction using Benjamini–Hochberg correction with a selected False Discovery Rate of 0.05 (see Supplementary Data 2). Contigs <5 kbp were excluded from this analysis.

We also compared gene presence/absence and genome region duplication or deletion between isolates from the same outbreak site and between isolates from different outbreak sites. To do this, we aligned the Illumina data from all time-series isolates (BundBos2013, BundBos2018, Luik2014, Luik2017, Rob2014, and Rob2015) to the Bundbos2013 assembly (elected as the most complete and contiguous *Bsal* assembly), and we also aligned the Illumina data from Luik2014 and Luik2017 to the Luik2014 assembly, and the Illumina reads for Rob2014 and Rob2015 to the Rob2015 assembly. All aligned bam files were processed, sorted, and duplicates removed using GATK as described in the "Variant calling" methods above. We then ran CNVnator[28] on each aligned bam file—this produces a list of candidate regions of duplication and deletion. We used bedtools[55] intersect to identify genes annotated within candidate deleted and duplicated regions and performed pairwise comparisons of (a) all isolates against the BundBos2013 assembly and (b) time-series pairs against the outbreak assembly, to assess the number, size, and gene content of regions differentially duplicated or deleted.

**Gene prediction, annotation, and repetitive regions.** Repetitive elements in the Falcon Unzip assemblies were identified using RepeatModeler v1.0.7 to identify

*de novo* repeats and RepeatMasker[51] v 4.0.5 to annotate them in the assembly; the output from RepeatModeler was combined with the known repeats in Repbase from *Batrachochytrium* species to create the repeat database input for Repeat-Masker. Structural variants were annotated using Assemblytics; thus alignments between all isolate assemblies and the BundBos2013 assembly were constructed with nucmer and advised -maxmatch -l 100 -c 500 parameters, and the delta files were input into the Assemblytics platform version 1.0 with default unique sequence length requirement, minimum variant size set to default 50, and maximum variant size set to 20,000.

Gene prediction and annotation were performed utilising a *de novo* eukaryotic fungal annotation pipeline. We generated a training set with RNA expression data from Farrer et al.[14] (BioProject PRJNA323392) for Braker v1.7, which predicts gene models using Genemark[56] v4.31 and Augustus[57] v3.2. Genes containing PFAM domains found in repetitive elements or overlapping tRNA/rRNA features were removed. Genes were named and numbered sequentially across each assembly contig in numerical order. PFAM domains[58], KEGG assignments, and Gene Ontology (GO) terms were annotated to genes of known function. PFAM (release 27) and TIGRFam (release 12) domains were identified using HMMER3[59]. GO terms were assigned using Blast2GO v2.3.5[60], with a minimum e-value of $1 \times 10^{-10}$. SignalP 4.0[61] and TMHMM[62] were used to identify secreted protein and transmembrane protein domains, respectively, the former has been associated with a high false-negative rate in basal chytrid fungi and so our SignalP secretion signal predictions may be an underestimation[63].

To explore the protease composition of the five isolates, we identified the top hit pairs from BLASTp searches (e-value <1e−5) against the MEROPS "pepunit.lib" database containing 1,103,662 proteases (downloaded from http://merops.sanger.ac.uk/ 22 April 2018). CBMs were identified using blastp (e-value <1e−50) top high scoring pair searches against the Carbohydrate-Active enZYmes Database CAZyDB.07202017 file (http://www.cazy.org/, downloaded 7 June 2018 containing 921,174 protein sequences). A list of CAZyme families, Pfam domains, and Kegg EC annotations associated with metabolism of plant material were curated based on fungal plant metabolic literature[20,21,64]. Enzymes potentially involved in the metabolism of plant biomass were identified based on Pfam and Kegg domain annotations and Blastp hits against the CAZyme family database.

Gene trees were constructed using RAxML v8.1.15 with 1,000 bootstrap replicates and nucleotide transition model GTRCAT and FastTree v2.1.8 from nucleotide sequences aligned using MUSCLE v3.8.31. Most comparisons of protein numbers were completed using only primary contig annotations (excluding alternative haplotypes) to avoid falsely inflating our gene counts and distortion due to unequal phasing of our assemblies. However, when considering the enzymatic capacity of isolates we also considered the alternative contigs: if a protein homologue was absent from the primary contigs we checked if it was present in the alternative contigs and included those protein counts in the analysis. Given that these represented single gene copies for a limited number of enzyme-encoding genes, they should not significantly distort our protein counts but hopefully reduces the influence of the random assignment of primary alternative contigs during the phasing of the assembly on our understanding of isolate function.

Potential Crinkler (CRN) genes were identified by blastp searches (e-value <1e−20) against the candidate CRN genes identified by Farrer et al.[14] in four chytrid species. Protein orthologs were identified between all isolate haplotypes using OrthoFinder[16] version 2.5.2 including *Bd* Jel423 (PRJNA13653, GCA_000149865.1) as an outgroup. Gene trees for orthogroups and candidate effector proteins were inferred from alignments generated in Muscle v 3.8.31 using RAxML v8.1.15 with 1000 bootstrap replicates and nucleotide transition model GTRCAT.

**Phylogenetic analysis.** We inferred the phylogenetic relationship of the *Bsal* isolates by aligning Illumina reads from all *Bsal* isolates, plus the *Bd* Jel423 isolate (PRJNA13653) as an outgroup, to the primary contigs of the BundBos2013 Falcon assembly, as the most complete and contiguous assembly, using BWA v0.7.12 mem command. As SNAPP requires a smaller subset of high confidence variants, we only included Illumina data for each isolate, as this technology has a lower error rate and established variant quality assignment and filtering techniques. We then performed variant calling and filtering as above, annotating 823,366 variants. Based on recommendations for usage of SNAPP from Beast2 (recommended for analysing smaller subsets of high quality SNPs), we further filtered using bcftools filters "AC == 0 ||AC == AN||F_MISSING > 0 ||QD < 20", leaving 111,051 SNPs, before filtering for only biallelic with no missing sites and linkage disequilibrium (LD) using SNPrelate[65] version 1.14.0 with an LD threshold of 0.7, retaining 8539 SNPs. We then performed a SNAPP version 1.42 analysis from BEAST version 2.6.0 with an estimated strict molecular clock, using tip-dating and accounting for the total invariant sites by adjusting the proportion.Invariant parameter. As per recommendations, we ran three independent SNAPP runs, with 60 M iterations (Markov chain Monte Carlo (MCMC)) each, combined using logCombiner and a 15% burn in, generating a total of just over 156 M iterations (MCMC). We compared log output files in Tracer v 1.7.1 to check estimates had converged before using treeAnnotator from BEAST2 v 2.6.0 with a 15% burn-in using the median heights to calculate node heights. We did not include *Bd* Jel423 as an outgroup as SNAPP cannot handle missing data and finding a subset of variants that discerned between the *Bsal* isolates but was conserved with the *Bd* genome resulted in a variant file with only a small number of contigs represented.

However, for other phylogenetic methods we included *Bd* Jel423 isolate as an outgroup, using Illumina reads from BioProject PRJNA13653, we performed the same SNP filters as above but allowing one missing group such that high-confidence SNPs missing in the *Bd* isolate could be included, and a greater proportion of the genome be represented. Thus, 36,495 SNPs were used for SNP-based PCA and Identity by Descent analyses in R[66] version 3.5 using SNPrelate[65] version 1.14.0 and plotted in ggplot2[67] from tidyr version 0.8.2. We also inferred phylogenies from this data set with RAxML v7.3.0 with 1000 bootstraps using the GTRCAT model of rate heterogeneity. We also inferred a core ortholog tree with OrthoFinder version 2.5.2, based on the trees of 4073 orthogroups containing genes from all *Bsal* isolates and the *Bd* Jel423 assembly (PRJNA13653, GCA_000149865.1). Initially, the principal and alternative contigs for the phased Luik2014 and BundBos2013 assemblies were treated as separate assemblies in this analysis; however as the principal and alternative contigs clustered to form a Luik2014 clade and a BundBos2013 clade, the analysis was simplified by including only the (more complete) principal contigs.

**HGT events**. Candidate genes acquired via HGT events were identified using two phylogenetic tools, with the candidates identified as very likely candidates by both tools being used as a higher-confidence candidate set. First, we ran DarkHorse version 2.0[68], for which the informative sequences for diamond BLASTp BLAST[49] v2.2.30 searches were generated from the NCBI nr database downloaded 16 August 2018 supplemented with the genomic data generated in this study. We excluded the taxonomic group Rhizophydiales *incertae sedis* (NICBI Taxonomic ID 1142503) and filtered for a minimum alignment length of 70% and filter thresholds of 0.02, 0.05, and 0.1; as recommended, we tried filter values <0.1 as at the time there were few closely related species sequenced. We then selected candidates with a normalised LPI (lineage probability index) value <0.2. Second, we also ran Alienness[69] v.1.0 to identify genes "very likely" of HGT origin based on an ai (Alien Index) value >30, inputting the Rhizophydiales taxonomic group as our sample's taxonomic group and the Chytridiomycetes class as our "group of interest" so looking for HGT not originating from this class, with input BLASTp BLAST[49] v2.2.30 searches blasting to the NCBI nr database accessed on 24 August 2018. The candidates identified by both techniques were taken as an initial candidate list. We tested this technique on the *Bd* data set from Sun et al.[70] as a function test and manually checked all candidates identified by our methods and by Sun et al.[70] by inspecting gene distance trees generated using pairwise BLAST alignments on the NCBI BLASTp platform against NR database (accessed 22 June 2021) and rejecting candidates that clustered in clades with a non-*Batrachochytrium* chytrid species or other closely related fungus where no other chytrid sequences were identified. Using these criteria, we rejected 37 of the 50 candidates identified by Sun et al.[70], retaining 13 confirmed HGT candidates; this may reflect both the addition of new chytrid assemblies to NCBI since 2016 and the use of the RefSeq database of NCBI by Sun et al.[70] (compared to the nr database used in our study)—with for example many candidates clustering with *Spizellomyces punctatus*, which was uploaded to NCBI in 2009 but not included in the RefSeq database. Our methods identified ten HGT candidates—of these, four were candidates identified by Sun et al.[70] that were confirmed as HGT candidates on manual gene tree inspection, one was a gene identified by Sun et al.[70] but rejected upon manual gene tree inspection, four were candidates not identified by Sun et al. (2016)[70] but confirmed as HGT candidates by manual gene tree inspection and one was a candidate not identified by Sun et al.[70] and rejected by on manual gene tree inspection.

Given the high number of HGT candidates in the Luik2014 isolate, we were concerned about possible contamination. However, all HGT candidates were located on contigs that contained other genes with fungal or chytrid genes as their closest match by BLAST and none of the contigs had BLAST hits to another species spanning >9.5% of the contig sequence. Of the Luik2014 HGT candidate genes, roughly 160 are spread along the 10 largest contigs of the assembly and the Luik2014 HGT candidate genes had top BLAST hits from >20 phyla rather than a single origin that would have suggested a contaminating species. Furthermore, the sequence coverage of the candidate HGT genes was also within the mean +/− one standard deviation, further supporting a single genome origin. Manual inspection of gene trees for a subset of candidates was consistent with HGT phylogenetic patterns.

**In vitro experiments**. Isolates were cultured in TGhL broth at 15 °C as per Martel et al.[9]. For all experiments, spores were collected in sterile distilled water. To remove mature sporangia, the aspirated mixture was passed over a sterile mesh filter of pore size 40 μm (Pluristrainer, PluriSelect). The flow through was centrifuged once at $3000 \times g$ for 5 min at 12 °C and the supernatant was removed. The pellet was resuspended in sterile distilled water and used as the zoospore fraction (>95% purity). The Catalan2018 isolate is not included in these experiments as it had not yet been isolated at the time of the experiments and subsequent genomic analyses placed this isolate within the spectrum of the isolates already tested.

**Saprotrophic growth experiments**. A vegetative saprotrophic environment was mimicked using a LB-based media, adapted from the "Baits and Culture Recipes" of the Maine Chytrid Laboratory website (https://umaine.edu/chytrids/isolation-methods-for-chytrids/baits-and-culture-media-recipes/). To allow greater control

of media concentration, the medium was not diluted to 1000 ml as per the Maine Chytrid Laboratory instructions but autoclaved at 400 ml and diluted with sterile distilled water to the desired concentration at the time of subculturing.

**Short-term comparison of isolate saprotrophic capacity**. Spore solutions were blinded and standardised to $5 \times 10^5$ spores in 1 ml sterile AD and added to 9 ml of vegetative-based media (4 ml of LB media with 5 ml of sterile AD) in 50 ml filter-top culture flasks. We included a positive control of 1 ml of the same spore solution in 9 ml of TGhL media (initially diluted 5 ml TGhL to 4 ml sterile AD to facilitate spore attachment). Flasks were incubated for 10 days at 15 °C and checked for growth and contamination daily; media was refreshed after 24 h (replacing dilute TGhL media with normal strength once spores had attached) and on day 5. On days 5 and 10, all flasks were visually scanned and three representative fields of vision at ×20 magnification were photographed and analysed for the number of mature sporangia, the number of motile spores, and sporangia coverage. As the reproductive body of *Batrachochytrium*, sporangia measures may be considered a stronger representation of growth than motile sporangia count—which in the time frame of this experiment could represent either (i) increased production of spores or (ii) reduction in attachment and development to sporangia. Growth was also tested using entire mixed culture solution, with a homogenised solution from scraped culture flasks suspended in the TGhL medium. This solution was standardised to a spore count of $1 \times 10^6$ spores/ml but showed varying numbers of sporangia (ratio of sporangia number BundBos2013:Luik2014:Captive2015-1:Rob2015:Captive2015-2 equal to 0.5:1:1:1:0.1). Once standardised by spore count, 1 ml of culture was added to 9 ml of LB medium or TGhL (for controls). Cultures were monitored daily for contamination and 3 representative ×20 field of visions were photographed at days 5 and 10. Three measures of growth: the number of motile spores, the number of sporangia and the proportion of coverage by mature sporangia, were calculated. Counting of LB and control cultures was blinded.

To assess the ability of *Bsal* isolates to continue to grow in the saprotrophic media in the long term, we selected the Luik2014 and BundBos2013 isolates as these showed significantly more promise of culture growth after 10 days in the LB medium. BundBos2013 and Luik2014 isolates were subcultured from cultures in TGhL broth using standard techniques. One millilitre of subculture was added to 9 ml of a media (see SI Table 10), performed in duplicate; media was refreshed every 4–5 days, and cultures were subcultured every 9–10 days. Isolates were observed daily, looking for motile spores and growth and maturation of sporangia.

Negative binomial generalised linear models were fit to spore and sporangia counts using the MASS[71] package with the structures (moving spore count ~ Isolate * Medium * Day) and (sporangia count ~ Isolate * Medium * Day), and the proportion of sporangia coverage was modelled using beta regression models from the betareg[72] package with structure (proportion of sporangia coverage ~ Isolate * Medium * Day), both in R[66] version 3.5.

**Complex plant material experiments**. Spores were collected from the five isolates as above and standardised to $1 \times 10^6$ spores/ml and the solutions blinded. In a 48-well culture plate, a $6–8 \times 1 \times 1$ mm piece of plant material (autoclaved commercial hay, pasteurised (heated in sterile AD to 70 °C for 1 h) commercial hay, untreated commercial straw, untreated commercial hay, or pasteurised beech leaf litter) was added to each well. These plant materials were selected based on the recommendations for baiting vegetative chytrids available on the Maine Chytrid Laboratory website (https://umaine.edu/chytrids/isolation-methods-for-chytrids/) together with beech leaf material representative of the terrestrial niche of European Fire Salamander populations. As a pilot study indicated the plant sample could influence attachment, we set up the plates ensuring each column of wells contained segments from the same plant sample to allow nesting of samples as a random effect in fixed effect models. To each row (8 wells) per plate, we added 500 μl of a blinded isolate spore solution ($5 \times 10^5$ spores per well). After incubating at 15 °C for 24 h, each plant material sample was carefully moved into a new well containing 500 μl of sterile distilled water. The number of mature sporangia with rhizoids attached to the plant material, visualised using light microscopy, was counted at days 4–5, 10, 15, 20, and 25. In early experiments, data were only collected at days 4 and 10, and in non-autoclaved material only data collected at day 4–5 was considered valid due to a high rate of bacterial contamination (see Supplementary Fig. 15).

For each experiment, a positive control (spore solution in TGhL medium) and two negative controls (plant sample in sterile distilled water and plant sample with heat-killed spores (heated for 1 h at 70 °C)) were included to check that (a) the spores had not been killed by the processing, (b) that there were no non-*Bsal* chytrids present or other structures that could be confused with *Bsal* sporangia and (c) that the attachment and maturation of sporangia was an active process. In total, across 6 experiments, 8 wells each of untreated hay, 8 wells of pasteurised leaf litter, 16 wells of pasteurised hay, and 48 wells containing autoclaved hay were analysed per isolate.

The experiment noted above was repeated 3 times using BundBos2013 and BundBos2018 isolates using only autoclaved hay, 12 samples per isolate per experiment and counting sporangia numbers at days 5, 10, 15, 20, and 25. The R packages glmmADMD[73] version 0.8.3.3 and glmmTMD[74] version 0.2.3 were used to fit mixed effect models to the data using the Random effects structure (Day|Experiment/plant_sample_id) to avoid temporal pseudo-replication and

account for the blocking of the effect of the plant sample within the experiment. A zero-inflated negative binomial model from glmmADMB package best fit the BundBos2013–BundBos2018 comparison (model structure Spore count ~ Day * Isolate + (Day|Experiment/sample), zeroInflation = TRUE). A truncated negative binomial hurdle from glmmTMB best fit growth on beech litter data (model structures: truncated negative binomial model of counts: Sporangia count ~ Isolate * Day + (Day|Experiment/sample), binary model of zero count best modelled with only Day as explanatory variable). A zero-inflated negative binomial model from package glmmTMB best fit the five isolate growth on hay data (Spore_number ~ Isolate * Day) + (Day | Experiment/Sample) with zero inflation model (Isolate + Experiment + sample). Model fit was accessed using the DHARMa package[75] version 0.3.4. The packages extrafont version 0.17 and patchwork version 1.1.1 were used for generating figures.

**Reporting summary**. Further information on research design is available in the Nature Research Reporting Summary linked to this article.

## Data availability

The raw sequence data, genome assemblies, and annotations have been deposited at GenBank as BioSamples SAMN14316973–SAMN14316981 and SAMN17104240–SAMN17104244, and SRA objects SRR11252126–SRR11252134 under BioProjects Accession numbers PRJNA610831, PRJNA623494, PRJNA623493, PRJNA623491, and PRJNA623492. Other data referenced in this paper are outlined in Supplementary Information. Other publicly available data used in this manuscript: MEROPS "pepunit.lib" database containing 1,103,662 proteases (downloaded from http://merops.sanger.ac.uk/ on 22 April 2018), Carbohydrate-Active enZYmes Database CAZyDB.07202017 file (http://www.cazy.org/, downloaded 7 June 2018 containing 921,174 protein sequences), the Bd Jel423 assembly (BioProject PRJNA13653, Accession GCA_000149865.1), the Bsal BundBos2013 assembly (BioProject PRJNA311566, Accession GCA_002006685.1), Bsal RNA expression data used for training assembly gene annotation (BioProject PRJNA323392), NCBI nr database used in HGT analysis was downloaded 16 August 2018 (available at https://ftp.ncbi.nlm.nih.gov/blast/db/nr*)

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

## Acknowledgements

M.K. was supported by FWO aspirant grant number 1111119N and would like to thank D. Gullick and JJADMS for their support in computational and practicalities.

## Author contributions

Study design: M.K., C.A.C., A.M., and F.P. Data analysis: M.K., C.A.C., T.P.S., J.F.M., S.C., and A.M. Performed and interpreted experiments: M.K., A.M., and F.P. Wrote the manuscript: M.K., all authors contributed to editing the manuscript.

## Competing interests

The authors declare no competing interests.
