## [Peer Review File · Nature Communications]

REVIEWER COMMENTS

Reviewer #1 (Remarks to the Author):

This study presents a comparative genomic analysis of several isolates of the amphibian-decline fungus, *Batrachochytrium salamandivorans* (Bsal), along with in vitro studies of these isolates saprobic potential. The authors find an astounding amount of genetic (gene and repeat content) diversity among their Bsal isolates that are temporally and geographically diverse. They document that isolates vary significantly in their gene content as related ability to degrade plant material, and embarked upon in vitro assays to examine these potential differences. Their work suggests that a couple of the Bsal isolates (Luik2014 and BundBos2018) have the ability to grow saprobically on plant tissue, including autoclaved beech leaves, native to Europe. This study provides the first evidence that some Bsal isolates may be able to survive between outbreak events by saprobically degrading plant material, and paves the way for future work examining how different isolates (with different ecological capabilities) may be contributing differentially to outbreak events. This study has huge implications toward management of the disease and the importance in preventing further introduction events (within and outside of Europe). It is also a good reminder that fungal pathogen populations need to be studied holistically, not just one isolate at a time.

Bigger Comments:

-Figure 1: Why no outgroup (e.g. Bd?) to root the tree? Would this not give you more insight into the evolution of these isolates and the spread of the epidemic? Also, you talk in the text about the conflicting topologies you recover. Why was this the topology that was chosen? I think an outgroup would have extremely helped this situation. This must be included.

-Data Availability. The authors seem to indicate that they have only deposited the raw data in the SRA. It is *very* frustrating to see that researchers are still only publishing their raw data instead of both raw data and assemblies and proteins with annotations (especially ones at top institutes with extensive genomics expertise!). This is non-transparent, un-replicable (due to differences in software availability, versions, etc.), and hinders future comparative studies outside of the authors' labs.

-The number of some of the orthologous clustering and identification of unique proteins is surprising which you explore somewhat in the results and discussion. Specifically in Table S3, it says Luik2014 has 11,965 unique proteins, but it only has ~17,000 proteins total. Is the suggestion that Luik2014 has lost many other genes that are being retained in the other strains, but has undergone lineage specific expansions (or lineage specific retentions of ancestral proteins) of a large number of its remaining proteins? I guess another hypothesis is that these genes are just diverged enough (in sequence) that they don't cluster with their orthologs (a hypothesis that could be bolstered by your staggering rates of sequence divergence you report later in the results section). Perhaps an analysis or presentation of the "core" set of proteins shared by all strains analyzed (I'm thinking a venn diagram might be easy). Based on the data tables presented I'm guessing the number of proteins shared between all strains is fairly low, maybe in the neighborhood of ~6,000 proteins or less, despite the fact that all strains have >10,000 proteins. How does this core set of proteins compare with Bd / other chytrids? It's discussed in a few specific examples, but not holistically. This could be explored more thoroughly.

Smaller Comments:

-Oxford commas needed throughout, including in title.

-Consistency of writing isolate names should be checked throughout (e.g. Captive2015 vs CAPTIVE2015 line 90).

-The number of some of the orthologous clustering and identification of unique proteins is surprising which you explore somewhat in the results and discussion. Specifically in Table S3, it says Luik2014

has 11,965 unique proteins, but it only has ~17,000 proteins total. Is the suggestion that Luik2014 has lost many other genes that are being retained in the other strains, but has undergone lineage specific expansions (or lineage specific retentions of ancestral proteins) of a large number of its remaining proteins? I guess another hypothesis is that these genes are just diverged enough (in sequence) that they don't cluster with their orthologs (a hypothesis that could be bolstered by your staggering rates of sequence divergence you report later in the results section). Perhaps an analysis or presentation of the "core" set of proteins shared by all strains analyzed (I'm thinking a venn diagram might be easy). Based on the data tables presented I'm guessing the number of proteins shared between all strains is fairly low, maybe in the neighborhood of ~6,000 proteins or less, despite the fact that all strains have >10,000 proteins. How does this core set of proteins compare with Bd / other chytrids? It's discussed in a few specific examples, but not wholistically. This could be explored more thoroughly.

-Figure S1 could be the beginning of a great review paper that many people would read and cite.

-Figure S3A&B title says "orthomcl single copy core ortholog phylogeny" but both the A and B legends say both trees are derived from SNPs?

- Line 96 and Figure 1: I am confused by the HPD range here. It says "58,416 years ago (95% HPD range 41-4.6x10⁸ years)." HPD is typically written as a range of years, why is the 108 included?

In-line Comments:

23: "per-outbreak" is an odd term to me, and might be worth defining here since you use throughout the paper and Nat Comm is a general audience journal.

55: "order" to "orders"

268: "saprotrophic lifestyle" to "saprotrophic potential" or "saprotrophic ability" – a distinction here just to clarify that you have not identified or observed this in the wild, but have documented this ability.

283-295: very nice discussion here.

362: BUSCO. For what it's worth, it might be worth checking out FGMP for future work – we feel it works better than BUSCO for fungal genomes (especially from under-sequenced lineages). Full disclosure, I was not a co-author of this paper, we've just found it useful.

<https://bmcbioinformatics.biomedcentral.com/articles/10.1186/s12859-019-2782-9>

794: "from an outbreak in..." missing end of that sentence. Where is it from?

Reviewer #2 (Remarks to the Author):

What are the major claims of the paper?

The authors present a nice bit of work describing the utility of genome sequencing and assembly on a per isolate basis to discover novel genomic traits associated with the emerging pathogenic chytrid fungus Bsal, a relative of Bd (causes of amphibian decline).

Are they novel and will they be of interest to others in the community and the wider field?

If correct, these findings are novel, and will be of interest to Bd researchers, but also to genomics field as a whole

Is the work convincing, and if not, what further evidence would be required to strengthen the conclusions?

I would like to see more information in table 1 and in the methods about the isolates. How were these obtained (clinical lab?) and identified in the first place, and from which species of amphibian? Why is there no location info for "captive" although states for privacy- I would think the country would be reasonable? Luik clade seems like a different species?

Figure 1. This tree needs an outgroup. Do the authors think these might in fact be different species? Jel423 mentioned in methods but is not on the tree (there is no gene tree that I could find).

Figure 2c/d these are basically unreadable and the legend is not very clear (what does the color legend mean? Is this with 2c?)

Figure 3- were stats performed on these data? If so, please indicate what is significantly different. 3F-figure legend seems to be missing some words (compared to what culture?)

Figure 4- total count is unnecessary. Luik2014 needs to be convincingly assessed to ensure no contamination or other issues.

There are a ton of data in the supplement, but very little context. Many are hard to read and legends are difficult to interpret. As examples- Fig S4: make the bars a different color- the outline is impossible! Fig S6- what does this even mean?

Line 419: gene prediction does not say versions used? Braker1? 2? Etc.

On a more subjective note, do you feel that the paper will influence thinking in the field?
Yes, if analysis proves to be correct

Please feel free to raise any further questions and concerns about the paper.
There are several minor grammatical errors and typos, and the paper should be carefully edited

Line 74 and 78: please state the number of isolates sequenced here

Line 228: maybe missing something here? Variation among strains that caused the epidemic?

I think this is potentially interesting, but the authors should hone the story and avoid presenting a great deal of confusing and poorly constructed figures. The main points that need to be addressed are

- 1) Is Luik clade a different species?
- 2) Are the captives and Catalan 2018 representative of introduced species?
- 3) Does BundBos/Rob clade represent resident Bsal

Unless I'm completely misinterpreting the data presented. There are no pathogenesis studies presented, and I think this is problematic for a pathogen. Also, clonality is so frequently stated, and yet I would think that cryptic recombination mechanisms are the only explanation for the results.

Reviewer #3 (Remarks to the Author):

In this manuscript Kelly et al. report de novo genome assemblies of six distinct *Batrachochytrium* salamandrorans outbreaks and their comparisons. They report significant differences between these genomes, which are interpreted as "outbreak-specific adaptations". Combining genome analysis and experiments they show some of these isolates are able of saprotrophic life-style. Understanding Bsal physiology and population genomics is very important to address this important animal pathogen that is devastating entire amphibian populations. This work provides important resources and some initial insights that need to be further developed. Although I consider this manuscript interesting and well performed, my main concern regards to the limitation of isolates analyzed (one or two representatives per outbreak) to sustain some of the claims made, and the lack of sufficient justification for the added value of getting a per-outbreak long-read assembly versus comparing a sufficient amount of isolates per outbreak using short-read technology and mapping to a single reference. Although the advantage of long-read based assemblies is undisputed for certain purposes such as the study of genomic rearrangements or transposable elements dynamics, these aspects do not constitute the main focus of this work. Gene presence/absence differences could be also discovered with less costly approaches.

These although perhaps less efficient and providing somewhat lower resolution would have the advantage of allowing for the analysis of many more strains per outbreak.

Main comments

line 85, assemblies are roughly described with some standard statistics. However to better assess completeness more information needs to be provided, such as number of Ns, how many contigs, whether these are likely to represent chromosomes, whether telomeric sequences are found, etc.

I was surprised to see that the reference genome for Bsal sequenced earlier by some of the authors (Farrer et. al. 2017) is not used for comparison, but some of the RNA data and other inferences are used.

Related to this, one of the selling points of this article is the advantage of using long-read based assemblies per outbreak versus mapping to a single genomic reference. To me this advantage is not shown. It is not clear whether the differences found here would have been missed by mapping to a single reference or with alternative methodologies. Short-read based assemblies can be very fragmented but can have very similar representation of the gene-space, also one could map to a reference and identify duplicated and lost genes, and by assembly of unmapped reads get an insight of isolate-unique genes. The "One reference – many isolates" approach could have provided the advantage of enabling the sequencing of further isolates per outbreak, which would address another concern of this analysis.

The study includes some pairs of isolates from the same outbreak, which are compared based on the presence of snps and indels. This shows that isolates from the same outbreak, although isolated in different years are similar at the sequence level. However, as these pairs of isolates are not compared at the level of gene presence/absence this tells us little about whether gene content is stable or not within an outbreak. Rates of sequence evolution and of gene gain and loss can be very different in plastic genomes. As the focus of this paper is on the differences between isolates at the gene content level, I do not consider there is sufficient evidence to assume the single isolate per outbreak is representative for the whole outbreak. Comparisons of gene content can be done even without a long-read assembly (see above) and should be performed to sustain this claim. The authors rightly conclude "Bsal genomes are highly variable" in genome size and gene content, but this is at odds with the taken assumption that they are stable within an outbreak.

Throughout the paper the authors use the word "isolate" when sometimes they may refer to "population", "outbreak" or "lineage". Please revise this, as it is confusing. Isolate or clone originates for one individual sampled from a population.

The phylogeny shown in Figure 1 is puzzling, as the position of the root breaks one of the populations. Placing of the root (which seems wrong, if we compare with the phylogenies shown in the supplementary figures) may have also affected the relative lengths of the branches and the time estimates.

Most of the genomic differences found concern the atypical characteristics of the Luik14 isolate, which is described to have almost double amount of genes, many HGT-derived genes and large family expansions. Given a single long-read assembly for this outbreak it is unclear whether this strain represents a rarity in this regard or is a true representative of the source population. In addition, although a "plethora of mechanisms for rapid evolution" is mentioned there is very superficial investigation of what mechanisms have been driving these differences in gene content (TE expansions, tandem duplications, introgression, etc).

Page 182. Given that the genetic diversity present in the source populations is unknown the

assumption that differences between serial isolates represent mutations accumulated in that period of time is flawed.

Minor comments

line 38, even if the genus name has already been cited, full species name should be provided at first mention of a species.

Line 90, 249. Mention Catalonia refers to a Spanish region this may be unknown to readers

Supplement Figure S2. Are there missing data? I cannot see a comparison of the two captive isolates, or they do not belong to the same outbreak?

Line 199. an underlying mechanism for what?

Line 315. "contrary to most pathogens", but typical of fungal pathogens

Line 502. And any additional one? 26/75 only informs about putative false negatives, but what about false positives?

Reviewer #4 (Remarks to the Author):

This is an interesting and well-composed paper with novel conclusions about an ecologically important pathogen. Using a combination of long-read and short read whole genome sequencing, the authors reveal previously hidden genetic and phenotypic variation in the salamander chytrid fungus *Bsal*. The implications of the authors' results are very significant. These results provide mechanistic insight into the origins of variation in pathogenicity among *Bsal* isolates, and furthermore lead to phenotypic evidence of a saprotrophic stage in the *Bsal* life cycle. These findings will be of interest to the emerging disease field as well as applied conservation biologists working to prevent further *Bsal* outbreaks globally. The evidence presented in this paper alters existing assumptions the community has taken for granted about this pathogen species, and will drive new research directions in the field.

Beyond the insights into the genomic basis of pathogenicity and saprotrophy in *Bsal*, these results also suggest that either *Bsal* has been present in Europe for longer than previously understood, introduced to Europe multiple times from a diverse source population(s), or a combination of these. This result coupled with the associated findings of functional pathogenic variation among isolates changes how we understand the nature of the current European *Bsal* outbreaks and underscores the importance of genomic surveillance along with existing biosecurity measures against batrachomyxozoid species.

Methodologically, the authors show the utility of comparing multiple de-novo assemblies over assembly to a single reference as traditionally performed in the field. This demonstration of methods will likely lead to improved study design in comparative genomics studies of related pathogens. The *Bsal* genome assemblies reported here are a major improvement on the single, previously published *Bsal* genome, and the raw data made publicly available through GenBank will be a valuable resource to future research endeavors. The methods described for obtaining a tip-calibration on the molecular clock rate using isolates from different time-points in the outbreaks are interesting and worth reporting as a comparative data point. However, I was pleased to see the authors use caution when interpreting molecular dating analyses providing a good explanation of the assumptions and uncertainty around this particular analysis.

Overall, I believe the results and conclusions presented here will be a valuable contribution to the Batrachochytrium genetics community and of interest to the wider disease evolution community at-large.

Reviewer #1:

R1.1

This study presents a comparative genomic analysis of several isolates of the amphibian-decline fungus, *Batrachochytrium salamandivorans* (*Bsal*), along with in vitro studies of these isolates saprobic potential. The authors find an astounding amount of genetic (gene and repeat content) diversity among their *Bsal* isolates that are temporally and geographically diverse. They document that isolates vary significantly in their gene content as related ability to degrade plant material, and embarked upon in vitro assays to examine these potential differences. Their work suggests that a couple of the *Bsal* isolates (Luik2014 and BundBos2018) have the ability to grow saprobically on plant tissue, including autoclaved beech leaves, native to Europe. This study provides the first evidence that some *Bsal* isolates may be able to survive between outbreak events by saprobically degrading plant material, and paves the way for future work examining how different isolates (with different ecological capabilities) may be contributing differentially to outbreak events. This study has huge implications toward management of the disease and the importance in preventing further introduction events (within and outside of Europe). It is also a good reminder that fungal

We thank the reviewer for their time and helpful comments.

	pathogen populations need to be studied holistically, not just one isolate at a time.	
R1.2	Figure 1: Why no outgroup (e.g. Bd ?) to root the tree? Would this not give you more insight into the evolution of these isolates and the spread of the epidemic? Also, you talk in the text about the conflicting topologies you recover. Why was this the topology that was chosen? I think an outgroup would have extremely helped this situation. This must be included.	We have now included Jel423 Bd as an outgroup in the phylogenies included in Figure 2 and Supplementary Figures 3A, and 3B. We had chosen the phylogeny inferred by SNAPP as this provided an estimation for the timing of divergence. We still include this SNAPP inferred phylogeny in the supplementary materials but have replaced the phylogenetic tree in the main Figure 1 with a phylogenetic tree inferred using RAxML and including the Bd Jel423 isolate as an outgroup. We found including the Bd Jel423 isolate did not alter the phylogenetic relationships within the Bsal isolates inferred by RAxML and Identity by Descent methods.
R1.3	Data Availability. The authors seem to indicate that they have only deposited the raw data in the SRA. It is *very* frustrating to see that researchers are still only publishing their raw data instead of both raw data and assemblies and proteins with annotations (especially ones at top institutes with extensive genomics expertise!). This is non-transparent, un-replicable (due to differences in software availability, versions, etc.), and hinders future comparative studies outside of the authors' labs.	We agree with the reviewer that transparency and reproducibility are key! The genome assemblies and annotations were uploaded to GenBank before submitting—as was always our intention and was specified in the reporting summary, apologies for not making this clear in the data availability section. We have revised the Data Availability section: “The raw sequence data, genome assemblies, and annotations have been deposited at GenBank under BioProject PRJNA610831, BioSamples SAMN14316973-SAMN14316981 and SAMN17104240- SAMN17104244, and SRA objects SRR11252126-SRR11252134. Other data referenced in this paper are outlined in the supplementary information.”

R1.4	The number of some of the orthologous clustering and identification of unique proteins is surprising which you explore somewhat in the results and discussion. Specifically in Table S3, it says Luik2014 has 11,965 unique proteins, but it only has ~17,000 proteins total. Is the suggestion that Luik2014 has lost many other genes that are being retained in the other strains, but has undergone lineage specific expansions (or lineage specific retentions of ancestral proteins) of a large number of its remaining proteins? I guess another hypothesis is that these genes are just diverged enough (in sequence) that they don't cluster with their orthologs (a hypothesis that could be bolstered by your staggering rates of sequence divergence you report later in the results section). Perhaps an analysis or presentation of the "core" set of proteins shared by all strains analyzed (I'm thinking a venn diagram might be easy). Based on the data tables presented I'm guessing the number of proteins shared between all strains is fairly low, maybe in the neighborhood of ~6,000 proteins or less, despite the fact that all strains have >10,000 proteins. How does this core set of proteins compare with Bd / other chytrids? It's discussed in a few specific examples, but not wholistically. This could be explored more	We agree with the reviewer that the diversity of the Luik2014 assembly is surprising and warrants further discussion. Extending the analysis, we indeed found that rapid rates of sequence divergence led to ortholog sequences sufficiently divergent that OrthoMCL did not recognise these as clusters. One strong indication for this was the considerably higher rate of ortholog identification using Blastp vs Blastn (e-value of 1e-5 and minimum identity coverage of 30%). Blastp identified a further 6,290 Luik2014 "unique" genes with hits to genes within our other Bsal assemblies, including 5,493 Luik2014 genes with hits from 5+ of our isolate assemblies; Blastn identified fewer than half these (2,744 Luik2014 "unique" genes showing hits to the Batrachochytrium gene sets, 2295 with hits to 5 or more of the Batrachochytrium isolates). We therefore tried running another ortholog clustering program with a Blastp based approach – namely OrthoFinder¹, which indeed assigned a much higher proportion of Bsal isolates to an orthogroup, as summarised below (see also updated supplementary Table 3): OrthoFinder identified 10,403 orthogroups, 5,155 include all Bsal isolates, 9,102 include 2+ Bsal isolates, with 6,312 Bsal genes not assigned to any orthogroup. OrthoMCL identified 10,567 orthogroups, 4,687 include all Bsal isolates, 8,657 include 2+ isolates, but with 15,472

thoroughly.

Bsal genes unassigned to any orthogroup.

The higher rate of gene assignment allowed OrthoFinder to infer a phylogeny based on 4,073 trees (core to all *Bsal* isolates and *Bd* Jel423), while the small number of single copy core orthologs output by OrthoMCL had hindered our inclusion of a gene sequence inferred phylogeny. Congruous with our SNP-based phylogenetic analyses, the phylogeny inferred by OrthoFinder also shows the Luik2014 isolate diverging early from the other *Bsal* isolates and the BundBos2013 diverging before the Captive2015 and Catalan2018 isolates, however in this protein-based phylogeny the Rob2015 isolate is positioned with the latter.

While OrthoFinder had a higher rate of gene assignment to orthogroups, there remained 7,927 genes from the Luik2014 isolate for which no ortholog was found in our other *Batrachochytrium* isolates. To check the validity of these gene calls we looked for potential orthologs in assemblies from other Chytridiomycota species (listed in Table S4), Blastp (identity >30%, e-value < 1e-5) identified hits for all except 2,111 genes. A further 507 of the genes have potential orthologs in other species within the NCBI nr database (Diamond Blastp search with minimum e-value < 1e-5 against NCBI nr database downloaded 16th August 2018), and a further 498 have functional annotations (Pfam, TIGRfam, GO term) identified by HMMER3.

This left 1,106 genes with no annotations or orthologs identified. We plotted the average length of these genes to see if they were short gene calls, and found that while the

do have a shorter average length (843 amino acids) compared to the average length of the other Luik2014 genes (1,572 amino acids); and show a condensed gene length distribution compared with all other Luik2014 predicted genes (see histogram below), none were shorter than 100 amino acids long:

Based on the improved rate of orthogroup assignment, we

elected to include the OrthoFinder ortholog analysis in the manuscript. Notably, the inferred phylogeny was largely congruous with phylogenies inferred using SNP based approaches, and the improved ortholog clustering still highlights the Luik2014 isolate as an outlier with regards to the number of isolate-specific orthogroups (16-60 times more isolate specific orthogroups) and candidate effector proteins, supporting our initial conclusions and discussion. Because of all of the various combinations for gene presence/absence between 8 annotations (seven *Bsal* isolates plus *Bd* Jel423), we unfortunately couldn't construct a clear Venn diagram to represent this data. We have however updated Figures 2 and S3, Table S3, and have adapted the Main text:

Lines 100-104 we have added description of the OrthoFinder inferred phylogeny: "Phylogenetic analysis inferred using core ortholog sequence by OrthoFinder¹ based on protein annotations from PacBio assemblies also place the Luik2014 isolate as an early diverging clade with the BundBos2013 isolate next to diverge; however this protein sequence based approach positions the Rob2015 isolate in a clade with the Catalan2018 and Captive2015 isolates (see SI Figure 3A)."

Lines 125 and 130-132 we have adapted the discussion of the ortholog clustering analysis: "Ortholog clustering indicated 909 orthogroups unique to the Luik2014 isolate, containing hundreds of candidate effector proteins without homologs recognised in other *Bsal* assemblies or *Bd* Jel423

		(SI Table 3).” We have adapted the Methods Lines 528-530: “Protein orthologs were identified between all isolate haplotypes using OrthoFinder version 2.5.2 including Bd Jel423 (PRJNA13653, GCA_000149865.1, Farrer et al 2017) as an out group.”
R1.5	Oxford commas needed throughout, including in title.	We have added these throughout
R1.6	Consistency of writing isolate names should be checked throughout (e.g. Captive2015 vs CAPTIVE2015 line 90).	Thank you for pointing this out, we have checked this throughout
R1.7	The number of some of the orthologous clustering and identification of unique proteins is surprising which you explore somewhat in the results and discussion. Specifically in Table S3, it says Luik2014 has 11,965 unique proteins, but it only has ~17,000 proteins total. Is the suggestion that Luik2014 has lost many other genes that are being retained in the other strains, but has undergone lineage specific expansions (or lineage specific retentions of ancestral proteins) of a large number of its remaining proteins? I guess another hypothesis is that these genes are just diverged enough (in sequence) that they don't cluster with their orthologs (a hypothesis that could be bolstered by your staggering rates of sequence	We have answered this comment in R1.4 above

	divergence you report later in the results section). Perhaps an analysis or presentation of the “core” set of proteins shared by all strains analyzed (I’m thinking a venn diagram might be easy). Based on the data tables presented I’m guessing the number of proteins shared between all strains is fairly low, maybe in the neighborhood of ~6,000 proteins or less, despite the fact that all strains have >10,000 proteins. How does this core set of proteins compare with Bd / other chytrids? It’s discussed in a few specific examples, but not wholistically. This could be explored more thoroughly.	
R1.8	Figure S1 could be the beginning of a great review paper that many people would read and cite.	We agree that it is an interesting and important disparity!
R1.9	Figure S3A&B title says “orthomcl single copy core ortholog phylogeny” but both the A and B legends say both trees are derived from SNPs?	Thank you for pointing out this typing error, we had initially intended to include a single copy core orthology phylogeny inferred by RAxML based on the genes identified as single copy core orthologs by OrthoMCL. However, the low rate of ortholog assignation resulted in a very small number of core orthologs being identified. However, as detailed in R1.4 we have updated our ortholog clustering using OrthoFinder, which identified more than five times more groups with orthologs in all isolates, and so Figure S3 has been adapted to include a core ortholog tree inferred by OrthoFinder.
R1.10	Line 96 and Figure 1: I am confused by the	Our apologies, this was a formatting error, the text reads:

	HPD range here. It says “58,416 years ago (95% HPD range 41-4.6x10 ⁸ years).” HPD is typically written as a range of years, why is the 108 included?	“58,416 years ago (95% HPD range 41-4.6x10 ⁸ years).”
R1.11	23: “per-outbreak” is an odd term to me, and might be worth defining here since you use throughout the paper and Nat Comm is a general audience journal.	Thank you for highlighting the ambiguity of this term, to avoid adding a definition to the abstract we removed the reference to “per-outbreak” assemblies here. The term is now introduced, with greater context, in the main text. Line 23 now reads: “We generated de novo genomic assemblies for six outbreaks of the emerging pathogen Batrachochytrium salamandrivorans (Bsal), to reveal the European epidemic currently ravaging amphibian populations to comprise multiple, highly divergent lineages demonstrating outbreak-specific adaptations and metabolic capacities.”
R1.12	55: “order” to “orders”	This has been changed
R1.13	268: “saprotrophic lifestyle” to “saprotrophic potential” or “saprotrophic ability” – a distinction here just to clarify that you have not identified or observed this in the wild, but have documented this ability.	This read “saprotrophic lifecycle” not “saprotrophic lifestyle”, but has been changed to “saprotrophic potential” to further clarify.
R1.14	283-295: very nice discussion here.	Thank you!
R1.15	362: BUSCO. For what it’s worth, it might be worth checking out FGMP for future work – we feel it works better than BUSCO for fungal genomes (especially from under-sequenced lineages). Full disclosure, I was not a co-author of this paper, we’ve just found it useful.	Thank you for the recommendation- we will certainly try it out in future projects!

	https://bmcbioinformatics.biomedcentral.com/articles/10.1186/s12859-019-2782-9	
R1.16	794: "from an outbreak in..." missing end of that sentence. Where is it from?	The isolate is from the Montnegre i el Corredor Natural Park in Catalonia in Spain, the text has been adapted to include this: "Catalan 2018 was isolated from an outbreak in the Montnegre i el Corredor Natural Park, Catalonia, Spain."
Reviewer #2:		
R2.1	What are the major claims of the paper? The authors present a nice bit of work describing the utility of genome sequencing and assembly on a per isolate basis to discover novel genomic traits associated with the emerging pathogenic chytrid fungus Bsal, a relative of Bd (causes of amphibian decline). Are they novel and will they be of interest to others in the community and the wider field? If correct, these findings are novel, and will be of interest to Bd researchers, but also to genomics field as a whole Is the work convincing, and if not, what further evidence would be required to strengthen the conclusions? I would like to see more information in table 1 and in the methods about the isolates. How were these obtained (clinical lab?) and identified	We thank the reviewer for their time and helpful comments. Please note, line numbers throughout this document refer to the manuscript file with track changes viewed in simple markup. We have added more details in to the Methods section about how the isolates are collected, Lines 370-376 now read "All isolates were collected from infected fire salamanders (Salamandra salamandra), or a marbled newt (Triturus marmoratus) for the Catalan2018 isolate, naturally infected during their respective outbreak. Isolates were collected using protocols as described in Martel et al. (2013); 1-2mm skin sections are dragged through agar medium to remove surface contamination, before being placed in TGhL (tryptone, gelatin hydrolysate and lactose) medium with 200 mg/L penicillin-G and 400 mg/L streptomycin sulphate antibiotics and incubated at 15 °C."

	in the first place, and from which species of amphibian? Why is there no location info for “captive” although states for privacy- I would think the country would be reasonable? Luik clade seems like a different species?	Specimens from captive populations were included with the agreement of anonymity for the owners of the captive populations. The authors feel that due to the limited number of collections with Fire salamanders in some European countries, identifying the country of the collection could allow speculation over the collector’s identity and threaten their anonymity.
R2.2	Figure 1. This tree needs an outgroup. Do the authors think these might in fact be different species? Jel423 mentioned in methods but is not on the tree (there is no gene tree that I could find).	As detailed in R1.2 we have added the Jel423 Bd isolate as an outgroup to our phylogenies. We do not believe the Luik isolates to be a different species as it clusters closely with other Bsal isolates in SNP and protein based phylogenies. Furthermore, clinically – in terms of symptoms and species susceptibility, and in phenotypic data from in vitro studies it falls within the range of other Bsal isolates and it is detected as Bsal with the sensitive duplex Bd-Bsal and simplex Bsal qPCR protocols using primers for Bsal 5.8S rRNA gene
R2.3	Figure 2c/d these are basically unreadable and the legend is not very clear (what does the color legend mean? Is this with 2c?)	We have improved the resolution of these figures, we have also now coloured the entire clade according to the isolate of sequence origin, added a title (“Gene trees- Isolate of origin”) to the colour legend which applies to both 2c and 2d and added clarification and further context to the caption, which now reads: “(c) FastTree inferred gene tree of candidate serine endopeptidases in the S8A MEROPS protease family, a family of secreted endopeptidases associated with nutrition, showing expansion in the Luik2014 isolate following isolate divergence (148 copies in Luik2014, indicated here with

		orange coloured clades and labels, compared to 4-6 copies in all other isolates, coloured according gene trees legend); (d) The FastTree gene tree of candidate Crinkler proteins, which by comparison have undergone large expansion and divergence within Bsal, the majority of which occurred before isolate divergence resulting in most clades containing sequence from numerous or all isolates, also coloured according to the Gene trees legend. Clustering Crinkler candidates using CDhit identified 142 clusters with 2+ proteins, with the largest cluster containing only 9 proteins from multiple isolates indicating limited isolate-specific expansion;”
R2.4	Figure 3- were stats performed on these data? If so, please indicate what is significantly different. 3F- figure legend seems to be missing some words (compared to what culture?)	The statistical analyses of these data are described in lines 148-161 and 171-183. We have added indication of statistical significance to Figure 3D, but for other subfigures the relationships indicated by the mixed effect models are too complex to clearly depict on the figure, but are described in detail in the above mentioned text. We are not sure what error in the figure legend you refer to, figure 3F is a photo image of sporangia attached to autoclaved hay, this was included to allow the reader to visualise how the chytrid fungi attach and grow on the plant material and so no comparative photo is included (there is extensive literature featuring images for Batrachochytrium species growing in culture medium). We have however checked and revised all figure legends to aim to better contextualise them as described in R2.3

R2.5	Figure 4- total count is unnecessary. Luik2014 needs to be convincingly assessed to ensure no contamination or other issues.	We agree with the reviewer that the total count for Luik2014 dominates the figure. We had included the total count in Figure 4 as not all HGT candidates are annotated as belonging to one of the columns depicted, and some candidates are annotated as belonging to multiple of the columns depicted (e.g. a transmembrane CAZYme contains both the TMHMM signal and is annotated as a candidate CAZYme). We have therefore removed total count from the figure but added this context to the Figure legend which now reads: "Figure 4. Variation in genes identified as candidates for horizontal gene transfer. Comparison of number of horizontal gene transfer candidate genes, determined using phylogenetic techniques, and their annotations. Given that not all HGT candidates were annotated within one of these groups, and some maybe be included in more than one of these categories (e.g. a protease Pfam and a Pfam otherwise not present within the assembly), thus the total HGT candidate counts do not equal the sum of the depicted columns: BundBos2013 (total 43 candidates), Luik2014 (509), Captive2015-1 (14), Captive2015-2 (16), Catalan2018 (16), Rob2015 (14)." We were also concerned about the degree of diversification observed in the Luik2014 assembly and performed substantial QC. Firstly, all our assemblies have been uploaded to GenBank

and passed their checks for contamination. Secondly, as noted on line 422 all assemblies were initially checked with GAEMR. This programme produces a number of plots for identifying contamination including a “blast bubbles” analysis where all contigs are compared to the NCBI nr database using BLAST and depicted as circles proportional to the size, coloured by the species of the most significant hit. For the Luik assembly we found that of 508 primary contigs, none had a best hit to a non-eukaryote species, 50% of the fungal hits were to a chytrid (also note that there are still relatively few chytrid species assemblies included in the nr database!), *Batrachochytrium* was the best genus hit overall, no other species showed a total hit sequence length > 20,000 bp or best hit for >8 contigs, average proportion of a contig covered by a non-chytrid best hit was 0.3%, the largest proportion of a contig covered by a non-chytrid best hit was 10%. We also checked contig GC content and coverage to look for aberrations.

Finally, as discussed in the HGT analysis methods lines 605-613: “all HGT candidates were located on contigs that contained other genes with fungal or chytrid genes as their closest sequence match by BLAST and none of the contigs had BLAST hits to another species spanning more than 9.5% of the contig sequence. Of the Luik2014 HGT candidate genes roughly 160 are spread along the 10 largest contigs of the assembly and the Luik2014 HGT candidate genes had top BLAST hits from more than 20

		phyla rather than a single origin that would have suggested a contaminating species. Furthermore, the sequence coverage of the candidate HGT genes was also within the mean +/- one standard deviation, further supporting a single genome origin. Manual inspection of gene trees for a subset of candidates was consistent with HGT phylogenetic patterns.”
R2.6	There are a ton of data in the supplement, but very little context. Many are hard to read and legends are difficult to interpret. As examples- Fig S4: make the bars a different color- the outline is impossible! Fig S6- what does this even mean?	We have adapted Figure S4 as suggested. Figure S6 is a gene tree of all M36 metalloprotease candidates from the Bsal isolates. As discussed on lines 126-128, this figure was included, along with Figures 2c and 2d, to illustrate that these expansions in protein families occurred at different times – while the S8A serine proteases (Figure 2c) show huge expansion after isolate divergence, with 148 copies in the Luik2014 isolate and only 4-6 copies in the other isolates; the crinkler proteins show expansion before isolate divergence with nearly all clades containing a representative from each Bsal isolate; and the M36 metalloproteases show expansion both before and after isolate divergence with many clades containing candidates from all isolates, but also considerable expansion in the BundBos2013 isolate which has 73-123 more M36 metalloprotease candidates than the other isolates- however these expansions are seen as duplications within each clade rather than massive expansion within single clades as seen in the S8A serine peptidases. As for Figures 2c and 2d (described in R2.3) we have now coloured the entire clade

		according to sequence of origin to hopefully aid the reader in making these comparisons. We have also revised the legends of Figures 2c, 2d and S6 to better highlight this context and comparison.
R2.7	Line 419: gene prediction does not say versions used? Braker1? 2? Etc.	We have added versions to this text
R2.8	On a more subjective note, do you feel that the paper will influence thinking in the field? Yes, if analysis proves to be correct Please feel free to raise any further questions and concerns about the paper. There are several minor grammatical errors and typos, and the paper should be carefully edited	We have read through and checked the manuscript thoroughly.
R2.9	Line 74 and 78: please state the number of isolates sequenced here	We have added the number of isolates here
R2.10	Line 228: maybe missing something here? Variation among strains that caused the epidemic?	We do not believe there is an error here, the full sentence here reads: "Through a combination of phylogenetic, genomic and phenotypic methods we identified vast variation within the European Batrachochytrium salamandrivorans (Bsal) epidemic."
R2.11	I think this is potentially interesting, but the authors should hone the story and avoid presenting a great deal of confusing and poorly	We appreciate there is a lot of supplementary materials, but we believe all of the figures provide useful contributions to the manuscript. We have revised the figures and legends as

	constructed figures.	described above in R2.2-R2.6.
R2.12	The main points that need to be addressed are 1) Is Luik clade a different species?	As discussed above in R2.2 we do not believe the Luik clade is a different species, there is both genotypic and phenotypic evidence to support that it also represents Batrachochytrium salamandrivorans .
R2.13	2) Are the captives and Catalan 2018 representative of introduced species?	We also do not believe there is any evidence to suggest that the Captive2015 isolates or the Catalan2018 are different species, clinically, genotypically and phenotypically these also appear to be Batrachochytrium salamandrivorans .
R2.14	3) Does BundBos/Rob clade represent resident Bsal	We do not find it entirely clear what the reviewer means by "resident Bsal ". There is evidence that Bsal is still present within the populations in these regions- the BundBos2018 isolate represents was isolated from one of the first fire salamanders seen at the index site for over 5 years after the mass mortality events of 2013, and this isolate clusters with the BundBos2013 isolate in genomic analyses, and so we hypothesise that Bsal had remained present, in either other amphibian host- or saprotrophic- reservoirs, in the interim. However, population dynamics (namely the high susceptibility of endemic fire salamanders and mass mortality events), and the absence of Bsal in neighbouring populations that were unaffected by mass mortality events ² , do not suggest that these Bsal isolates were endemic to these populations.
R2.15	Unless I'm completely misinterpreting the data presented. There are no pathogenesis studies	As noted in the text lines 75 and 316-318, all isolates were associated with morbidity and mortality in the outbreak site

presented, and I think this is problematic for a pathogen. Also, clonality is so frequently stated, and yet I would think that cryptic recombination mechanisms are the only explanation for the results.

(wild or captive), and all isolates have been shown to cause fatal chytridiomycosis in infection trials settings.

So far infection trials testing *Bsal* isolates have never found a significant difference in mortality – e.g. Stegen et al (2017) include infection trial data from four of the isolates included in this paper and found no significant difference. These trials included very small sample sizes (e.g. 5 individuals per treatment), and with lower initial *Bsal* doses some (non-significant) differences in virulence were observed; a power analyses suggested that to differentiate such differences larger sample sizes of 20+ animals per group would be required. Getting ethical approval to perform such experiments (~200 animals for 9 isolates included in this study) can be difficult, and we determined this wasn't justifiable for this paper given that all isolates included have been shown to cause rapid mortality – both in the field and in laboratory conditions. Preliminary data suggests such experiments would be detecting nuanced differences in virulence, that we feel could prove insignificant in the context of a European outbreak.

There were two references to clonality within the manuscript, these have been removed, and greater discussion of recombination has been added as discussed in R3.8.

Reviewer #3:

R3.1	In this manuscript Kelly et al. report de novo genome assemblies of six distinct Batrachochytrium salamandrivorans outbreaks and their comparisons. They report significant differences between these genomes, which are interpreted as “outbreak-specific adaptations”. Combining genome analysis and experiments they show some of these isolates are able of saprotrophic life-style. Understanding Bsals physiology and population genomics is very important to address this important animal pathogen that is devastating entire amphibian populations. This work provides important resources and some initial insights that need to be further developed. Although I consider this manuscript interesting and well performed, my main concern regards to the limitation of isolates analyzed (one or two representatives per outbreak) to sustain some of the claims made, and the lack of sufficient justification for the added value of getting a per-outbreak long-read assembly versus comparing a sufficient amount of isolates per outbreak using short-read technology and mapping to a single reference. Although the advantage of long-read based assemblies is undisputed for certain purposes such as the study of genomic rearrangements or transposable elements dynamics, these aspects do not constitute the main focus of this work. Gene presence/absence differences could be also	We thank the reviewer for their very helpful comments. While we agree with the reviewer that additional isolates from each outbreak site would provide more data points for analysis, our sample size was not limited by the decision to use long read sequencing data and the increased sequencing/computation costs associated with this. The three pairs of isolates sequentially isolated from the BundBos, Luik and Rob outbreaks were, and remain to the authors’ knowledge, the only sequential isolations from Bsal outbreaks globally. This is because isolating Bsal from wild outbreaks is inherently difficult-  • At wild outbreak sites, Bsal has reduced host population size to <1% within 6 months of introduction³ so there is very quickly very few individuals left (the BundBos2018 isolate was isolated from one of the first fire salamanders seen at the index site in the five years since the initial mass mortality events). In captive populations, remaining individuals are always treated to eliminate Bsal from the population. • Bsal kills its host very quickly (2-3 weeks), and the period where individuals are exhibiting symptoms but still active is considerably shorter. The most common amphibian host species in the European epidemic, fire salamanders, are nocturnal and naturally cryptic – we can only find these animals when they are active
-------------	--	---

discovered with less costly approaches. These although perhaps less efficient and providing somewhat lower resolution would have the advantage of allowing for the analysis of many more strains per outbreak.

which requires a relatively narrow temperature range with a very high humidity. Thus, to find an infected individual there must be a night with these favourable weather conditions (some years there are only 20 such nights per year) during the very short period (perhaps only days) when the individual is exhibiting symptoms and still active. Furthermore, the conditions most favourable for host amphibian activity is also most favourable for *Bsal* growth and so individuals die quickest during these seasons.

- *Bsal* is an intracellular pathogen that is quickly outcompeted *in vitro* by bacterial, and other fungal, species – a successful isolation relies upon having a sufficiently high *Bsal* load with no or little other contamination- this is nearly impossible given that the late stage of disease is associated with loss of the skin integrity and overwhelming cutaneous infections and septicaemia. We frequently find of 400+ wells plated during an isolation perhaps 1-5 may have sufficient *Bsal* present to establish an *in vitro* isolate colony without contamination.

Combined, these factors mean there are only a few isolates of *Bsal* in culture- this manuscript represents the largest collection of *Bsal* isolates to date.

We agree with the reviewer that long and short read data have different advantages – which is why we combined analyses of both in this manuscript- using long read sequencing for our assemblies (which will support future

		work on analyses such as recombination and transposable elements), and short read data for our SNP-based analyses due to their high accuracy and better established variant identification and filtering pipelines.
R3.2	line 85, assemblies are roughly described with some standard statistics. However to better assess completeness more information needs to be provided, such as number of Ns, how many contigs, whether these are likely to represent chromosomes, whether telomeric sequences are found, etc.	The assembly summary statistics (including e.g. number of contigs) were presented in supplementary Table S1, we have added the number of gaps to this table. Unfortunately, none of our assemblies are at chromosome level, but they do none-the-less represent a substantial improvement in completeness compared to the previously published assembly (see R3.4 for comparison table)
R3.3	I was surprised to see that the reference genome for Bsal sequenced earlier by some of the authors (Farrer et. al. 2017) is not used for comparison, but some of the RNA data and other inferences are used.	We compare our assemblies to this previously published assembly (PRJNA311566) in:  • Lines 106-108 we discuss the size of the assemblies compared to the previously published assembly, • Lines 303-305 we compare contiguity, gene counts and discuss that long reads enabled the better resolution of repetitive elements (16.2Mbp in our BundBos2013 assembly vs 5.9Mbp in the previous BundBos2013 assembly). • Table S1 – we compare the number of contigs, the assembly length, the contig N50 and the GC of our assemblies and the previously published assembly • Table S8 – we compare gene counts and BUSCO scores of the previous assembly (10,138 genes,

		BUSCO score 71.3%) to scores from our assemblies (10,482-17,091 genes, BUSCO scores 91-97%) We decided such comparisons indicate our long read assembly for the BundBos2013 isolate to be an improvement and so included this genome assembly in our further analyses.
R3.4	Related to this, one of the selling points of this article is the advantage of using long-read based assemblies per outbreak versus mapping to a single genomic reference. To me this advantage is not shown. It is not clear whether the differences found here would have been missed by mapping to a single reference or with alternative methodologies. Short-read based assemblies can be very fragmented by can have very similar representation of the gene-space, also one could map to a reference and identify duplicated and lost genes, and by assembly of unmapped reads get an insight of isolate-unique genes. The “One reference – many isolates” approach could have provided the advantage of enabling the sequencing of further isolates per outbreak, which would address another concern of this analysis.	We agree with much of the content of this comment. We agree that short-read alignment based approaches can provide a lot of useful information, which is why we have used short-read alignment based approaches for our SNP-based analyses, and as described in R 3.5 we have now included such gene presence-absence analyses in the revised version of the manuscript. It appears that one of the main reservations the reviewer articulates about our approach of using long read data for de novo assembling is that it has imposed limits on the number of isolates, and particularly isolates sequentially isolated from the same outbreak sites, included in the analyses. This is a misunderstanding, which we have tried to clarify in this response (also see R3.1) and the new manuscript. Our analyses are not limited by the use of long read sequencing – our study includes all Bsal isolates for which genomic data had been successfully isolated, and represents the most complete collection of Bsal isolates to date. We have not elected to use long read sequencing over

including further isolates, and we do not try to argue that long read sequencing should be selected over including more isolates. We recognise the limitations of our sample size, and repeatedly refer to this throughout the discussion.

We do feel, particularly with these highly repetitive fungal genomes, that using long read sequencing improved our assemblies – as seen when comparing the long read BundBos2013 isolate assembly with the previously published assembly from Illumina data:

	PacBio phased BundBos2013	Illumina BundBos2013 (Farrer et al.2017)
Primary contig assembly length	41.6Mbp	32.6Mbp
Number of contigs	227	5,358
N50	346.3kbp	10.5kbp
Repetitive content	5.9Mbp	16.2Mbp
Gene count	10,138	12,269
BUSCO score	71.3%	96.6%

This data is all presented in the manuscript as detailed in R3.3.

Furthermore, while duplications and losses can be identified with a single reference, as the reviewer notes, lineage or isolate specific genes will be missed. Furthermore, as chytrid assemblies have been negatively affected by the high heterozygosity and repeat content, assembling multiple

		genomes provides support for gene identification in the species as some regions may not be assembled for all genomes.
R3.5	The study includes some pairs of isolates from the same outbreak, which are compared based on the presence of snps and indels. This shows that isolates from the same outbreak, although isolated in different years are similar at the sequence level. However, as these pairs of isolates are not compared at the level of gene presence/absence this tells us little about whether gene content is stable or not within an outbreak. Rates of sequence evolution and of gene gain and loss can be very different in plastic genomes. As the focus of this paper is on the differences between isolates at the gene content level, I do not consider there is sufficient evidence to assume the single isolate per outbreak is representative for the whole outbreak. Comparisons of gene content can be done even without a long-read assembly (see above) and should be performed to sustain this claim. The authors rightly conclude “Bsal genomes are highly variable” in genome size and gene content, but this is at odds with the taken assumption that they are stable within an outbreak.	We agreed that it would be informative to include gene presence/absence comparisons for pairs of isolates from the same outbreak site for pairs of isolate. We have incorporated this into the main text at lines 220-245, which now read: “To see if gene presence or absence data suggested greater stability in gene arsenal within an outbreak than between outbreaks we identified candidate regions of deletion or duplication using CNVnator²⁸. To allow comparisons between isolates from the same outbreak site versus isolates from different outbreak sites, we ran CNVnator on aligned bam files generated by aligning Illumina reads from all isolates to the BundBos2013 assembly, as well as aligning the Luik2014 and Luik2017 isolates to the Luik2014 assembly, and the Rob2014 and Rob2015 isolates to the Rob2015 assembly. We then performed pairwise comparisons of the size, and number of affected genes, of regions identified as deleted or duplicated in one of the isolate pairs (see Methods for more details, and Supplementary Figure 10). We found that the length of differentially deleted genome sequence in pairwise comparisons was smaller in isolate pairs from the same outbreak site than between outbreak sites (T-test $t=-2.873$, $df = 33.9$, $p=0.0069$, on average 312kbp differentially deleted in pairs from the same outbreak site versus an average of 487kbp differentially deleted in pairs from different outbreak sites). The number of genes deleted were not significantly higher or lower

between pairs from the same outbreak site compared to pairs from different outbreak sites (T-test $t=-0.93128$, $df=17.2$, $p=0.36$, on average 198 genes differentially deleted in pairs from the same outbreak site versus on average 245 genes differentially deleted in pairs from different outbreak sites). However, both the size of differentially duplicated sequence and the number of differentially duplicated genes were found to be smaller in isolate pairs from the same outbreak site compared to pairs containing isolates from different outbreak sites (T test duplicated region sequence size $t = -3.2644$, $df = 25.91$, $p\text{-value} = 0.003079$, gene count $t = -2.3485$, $df = 19.961$, $p\text{-value} = 0.02926$; mean 607kbp and 282 genes differentially duplicated in pairs from the same outbreak site, mean 1.54Mbp and 578 genes differentially duplicated in pairs different outbreak sites). Thus, while we see substantial variation within an outbreak site, it appears there is more diversification in the gene arsenal between isolates from different outbreak sites than isolates from the same outbreak site. "

We depict the results of this analysis in supplementary Figure 10, included below:

Pair wise comparison of differentially deleted or duplicated sequence

We describe the analysis in the method section, lines 472 - 480:

“We also compared gene presence/absence and genome region duplication or deletion between isolates from the same outbreak site and between isolates from different outbreak sites. To do this we aligned the Illumina data from all time-series isolates (BundBos2013, BundBos2018, Luik2014, Luik2017, Rob2014 and Rob2015) to the Bundbos2013 assembly (elected as the most complete and contiguous *Bsal* assembly), and we also aligned the Illumina data from Luik2014 and Luik2017 to the Luik2014 assembly, and the Illumina reads for Rob2014 and Rob2015

		to the Rob2015 assembly. All aligned bam files were processed, sorted, and duplicates removed using GATK as described in the Variant Calling methods above. We then ran CNVnator ²⁸ on each aligned bam file – this produces a list of candidate regions of duplication and deletion. We used bedtools ⁵⁴ intersect to identify genes annotated within candidate deleted and duplicated regions, and performed pairwise comparisons of (a) all isolates against the BundBos2013 assembly and (b) time-series pairs against the outbreak assembly, to assess the number, size and gene content of regions differentially duplicated or deleted.”
R3.6	Throughout the paper the authors use the word “isolate” when sometimes they may refer to “population”, “outbreak” or “lineage”. Please revise this, as it is confusing. Isolate or clone originates for one individual sampled from a population.	We have reviewed every use of the word “isolate”.
R3.7	The phylogeny shown in Figure 1 is puzzling, as the position of the root breaks one of the populations. Placing of the root (which seems wrong, if we compare with the phylogenies shown in the supplementary figures) may have also affected the relative lengths of the branches and the time estimates.	As described in R1.2. We have changed this figure to include the Bd Jel423 isolate as an outgroup in our phylogeny.
R3.8	Most of the genomic differences found concern the atypical characteristics of the Luik14 isolate, which is described to have almost double amount of genes, many HGT-derived genes and large family expansions. Given a single	We agree that our sample size, and the absence of any Bsal isolates from Asia, where it is thought to be endemic, makes it impossible to determine whether the variation seen in the Luik2014 isolate is an outlier or representative of endemic/source Bsal populations; which is why we do not

long-read assembly for this outbreak it is unclear whether this strain represents a rarity in this regard or is a true representative of the source population. In addition, although a “plethora of mechanisms for rapid evolution” is mentioned there is very superficial investigation of what mechanisms have been driving these differences in gene content (TE expansions, tandem duplications, introgression, etc).

try to make any claims in either direction. We only discuss the Luik2014 assembly as an outlier when compared to other assemblies from the European outbreak.

We had performed structural variant analyses (indeed one motivation for using long read sequencing data was to better facilitate such analyses). We also worked on further annotating transposable elements. During the process of writing the manuscript we had excluded these analyses, partially for concern about the volume of analyses and data in our supplementary materials (as commented on by Reviewer 2). However, in response to this comment we have integrated this within the discussion of repetitive elements and the “two speed” genome evolution in the supplementary materials, which now reads:

“Long read sequencing can also facilitate identification and analysis of larger structural variants (SVs). We used Assemblytics⁵ to identify structural variants of size 50-20,000 bp in isolates Luik2014, Captive2015-1, Captive2015-2, Rob2015 and Catalan2018 when aligned to the BundBos2013 assembly. This identified few such variants, with a maximum of 0.3% of an assembly identified as structural variants (in the Catalan2018) when compared to the BundBos2013 assembly. The fragmented nature of our assemblies already led us to assume that such techniques would not identify all structural variants; the conflicts experienced during phylogenetic analysis (see Main text and methods for discussion) indicate possible cryptic recombination, but these are hard to identify from

significantly fragmented assemblies. Furthermore, we found the degree of transposable element (TE) expansion in the BundBos2013 and Luik2014 isolates precluded genome alignment in TE heavy regions thus inhibiting the use of such alignment-based methods for SV identification. For example, the nucmer alignment used for Assemblytics analysis of the Captive2015-2 assembly compared to the BundBos2013 assembly (see Methods for details) indicated 32.9Mbp of alignments to 227 of the 233 BundBos2013 contigs, within which Assemblytics identified no minimal tandem (0bp) or repeat (6537bp) contractions. However, 2.4Mbp of the 8.8Mbp (27%) of the BundBos2013 assembly with no alignment to the Captive2015-2 isolate is annotated as either LTR or LINE TEs, compared to 1.5Mbp (4%) of the aligned regions. The Luik2014 assembly also seems to contain these TE expansions, as we see the inverse pattern of TE content in aligned vs unaligned regions (6% of unaligned regions as LINE/LTRs vs 12% of aligned regions), here unaligned regions do not seem repeat-rich in general, with 12.4Mbp of unaligned regions containing 3.8Mbp of repeats - below the overall proportion of repetitive elements in the BundBos2013 assembly (see Table7)."

We have added reference to these analyses in the main text at lines 252-260 which now read:

"There is evidence of transposable element (TE) expansion in both the BundBos2013 and Luik2014 isolate, with 4.8Mbp and 5.9Mb respectively annotated as LTRs and LINEs (compared to e.g. 732kbp in the Captive2015-2 isolate) and gene annotations indicating expansions in protease families

		associated with TEs with for example 253 BundBos2013 and 206 Luik2014 genes annotated as MEROPs A02X or A11X peptidase families, compared to 13-78 genes in the other Bsal isolate assemblies. However, we found little evidence of effector protein duplications and expansions linked with islands of low GC and highly repetitive content as hypothesised by the “two speed” genome (for further discussion see Supplementary Information).”
R3.9	Page 182. Given that the genetic diversity present in the source populations is unknown the assumption that differences between serial isolates represent mutations accumulated in that period of time is flawed.	We had tried to acknowledge the limitation of the unknown initial genetic diversity interpreting the rates of evolution with e.g. “...some of this divergence may have occurred before sampling within these clonal lineages, these estimates likely represent an overestimate of the long-term mutation”. We have changed the text here (lines 200-205) to more explicitly read “However, as this timescale is unlikely to allow for the effective removal of deleterious mutations by selective pressure, and the source population diversity remains unknown, with some of this divergence potentially occurring before sampling within these lineages, these estimates likely represent an overestimate of the long-term mutation rate⁶”
R3.10	line 38, even if the genus name has already been cited, full species name should be provided at first mention of a species.	This has been changed
R3.11	Line 90, 249. Mention Catalonia refers to a Spanish region this may be unknown to readers	This has been added

R3.12	Supplement Figure S2. Are there missing data? I cannot see a comparison of the two captive isolates, or they do not belong to the same outbreak?	The two Captive2015 isolates belong to outbreaks in two separate captive populations, this has been clarified in the caption for supplementary Figure S2, which now reads: "B) Principle Component Analysis of SNP variants for all isolates- isolates of the same colour are collected from the same site and seen to cluster together, Captive 2015-1 and Captive2015-2 isolates were collected from outbreaks in two separate captive populations"
R3.13	Line 199. an underlying mechanism for what?	Here we hypothesise that chromosomal aneuploidy could be the underlying mechanism for the copy number variation observed, as opposed to duplication of limited genomic regions.
R3.14	Line 315. "contrary to most pathogens", but typical of fungal pathogens	This is a good point, and we have changed the text to "contrary to most vertebrate pathogens,". While we of course agree that having an environmental reservoir is not uncommon in fungal pathogens; the scope of Batrachochytrium species to, so rapidly, drive populations to extinction is still an outlier; and we believe for the general readership of this journal it is still worth outlining the role of an environmental reservoir in allowing such fungal pathogens to divorce pathogen and host population densities.
R3.15	Line 502. And any additional one? 26/75 only informs about putative false negatives, but what about false positives?	We agreed that this was not adequately described and we had actually made an error in reporting this as Sun et al. (2016) ⁷ only reported 50 candidates for the Jel423 Bd isolate, we thank the reviewer for pointing this out so that we could correct it.

We have improved upon these comparisons by a) rerunning this analysis using the updated NR databases following the addition of a number of Chytridiomyces assemblies (so matching the database with which the *Bsal* assemblies were analysed), b) manually checking gene trees, inferred using pairwise blast comparison distance trees on the NCBI Blast platform, for all candidates identified by Sun et al. (2016)⁷ and by our HGT identification method. Candidates nested in a clade with non-*Batrachochytrium* chytrid species were rejected as HGT candidates. We thus rejected 37/50 candidates identified by Sun et al. (2016)⁷. Our methods identified 10 HGT candidates- of these 4 were candidates identified by Sun et al. (2016)⁷ that were confirmed as HGT candidates on manual gene tree inspection, 1 was a gene identified by Sun et al. (2016)⁷ but rejected upon manual gene tree inspection, four were candidates not identified by Sun et al. (2016)⁷ but that we confirmed as HGT candidates by manual gene tree inspection and 1 was a candidate not identified by Sun et al. (2016)⁷ and that we rejected by on manual gene tree inspection. We have adapted the text to reflect this –

Lines 587-603 now read: “We tested this technique on the *Bd* dataset from Sun et al. (2016)⁷ as a function test and manually checked all candidates identified by our methods and by Sun et al. (2016)⁷ by inspecting gene distance trees generated using pairwise BLAST alignments on the NCBI BLASTp platform against NR database (accessed 22nd June 2021) and rejecting candidates that clustered in clades with a non-*Batrachochytrium* chytrid species or other closely related fungus where no other chytrid sequences were identified. Using these criteria we rejected 37 of the 50 candidates identified by Sun et al. (2016)⁷, retaining 13 confirmed HGT candidates; this may reflect both the

		addition of new chytrid assemblies to NCBI since 2016, and also the use of the RefSeq database of NCBI by Sun et al. (2016)⁷ (compared to the nr database used in our study) – with for example many candidates clustering with Spizellomyces punctatus which was uploaded to NCBI in 2009 but not included in the RefSeq database. Our methods identified 10 HGT candidates- of these 4 were candidates identified by Sun et al. (2016)⁷ that were confirmed as HGT candidates on manual gene tree inspection, 1 was a gene identified by Sun et al. (2016)⁷ but rejected upon manual gene tree inspection, four were candidates not identified by Sun et al. (2016)⁷ but that confirmed as HGT candidates by manual gene tree inspection and 1 was a candidate not identified by Sun et al. (2016)⁷ and rejected by on manual gene tree inspection.”
Reviewer #4:		
R4.1	This is an interesting and well-composed paper with novel conclusions about an ecologically important pathogen. Using a combination of long-read and short read whole genome sequencing, the authors reveal previously hidden genetic and phenotypic variation in the salamander chytrid fungus Bsal. The implications of the authors' results are very significant. These results provide mechanistic insight into the origins of variation in pathogenicity among Bsal isolates, and furthermore lead to phenotypic evidence of a saprotrophic stage in the Bsal life cycle. These findings will be of interest to the emerging	We thank the reviewer for reading our manuscript so carefully, and thank them for their positive feedback.

disease field as well as applied conservation biologists working to prevent further *Bsal* outbreaks globally. The evidence presented in this paper alters existing assumptions the community has taken for granted about this pathogen species, and will drive new research directions in the field.

Beyond the insights into the genomic basis of pathogenicity and saprotrophy in *Bsal*, these results also suggest that either *Bsal* has been present in Europe for longer than previously understood, introduced to Europe multiple times from a diverse source population(s), or a combination of these. This result coupled with the associated findings of functional pathogenic variation among isolates changes how we understand the nature of the current European *Bsal* outbreaks and underscores the importance of genomic surveillance along with existing biosecurity measures against batrachochytrid species.

Methodologically, the authors show the utility of comparing multiple de-novo assemblies over assembly to a single reference as traditionally performed in the field. This demonstration of methods will likely lead to improved study design in comparative genomics studies of related pathogens. The *Bsal* genome assemblies reported here are a major improvement on the single, previously published *Bsal* genome, and the raw data made publicly

	available through GenBank will be a valuable resource to future research endeavors. The methods described for obtaining a tip-calibration on the molecular clock rate using isolates from different time-points in the outbreaks are interesting and worth reporting as a comparative data point. However, I was pleased to see the authors use caution when interpreting molecular dating analyses providing a good explanation of the assumptions and uncertainty around this particular analysis. Overall, I believe the results and conclusions presented here will be a valuable contribution to the Batrachochytrium genetics community and of interest to the wider disease evolution community at-large.	
--	---	--

NB. reference numbers in the quoted text above do not match the reference numbers in the manuscript to allow the generation of a clearer bibliography in this document. Please also note, line numbers throughout this document refer to the manuscript file with track changes viewed in simple markup.

References

1. OrthoFinder: solving fundamental biases in whole genome comparisons dramatically improves orthogroup inference accuracy | Genome Biology | Full Text.

<https://genomebiology.biomedcentral.com/articles/10.1186/s13059-015-0721-2>.

2. Sluijs, A. S. der *et al.* Post-epizootic salamander persistence in a disease-free refugium suggests poor dispersal ability of *Batrachochytrium* salamandrivorans. *Scientific Reports* **8**, 3800 (2018).

3. Stegen, G. *et al.* Drivers of salamander extirpation mediated by *Batrachochytrium salamandrivorans*. *Nature* **544**, 353–356 (2017).
4. Abyzov, A., Urban, A. E., Snyder, M. & Gerstein, M. CNVnator: an approach to discover, genotype, and characterize typical and atypical CNVs from family and population genome sequencing. *Genome Res* **21**, 974–984 (2011).
5. Nattestad, M. & Schatz, M. C. Assemblytics: a web analytics tool for the detection of variants from an assembly. *Bioinformatics* **32**, 3021–3023 (2016).
6. Ho, S. Y. W. *et al.* Time-dependent rates of molecular evolution. *Molecular Ecology* **20**, 3087–3101 (2011).
7. Sun, B. *et al.* Contribution of Multiple Inter-Kingdom Horizontal Gene Transfers to Evolution and Adaptation of Amphibian-Killing Chytrid, *Batrachochytrium dendrobatidis*. *Front. Microbiol.* **7**, (2016).

REVIEWER COMMENTS

Reviewer #1 (Remarks to the Author):

I am satisfied with the changes made by the authors to address the concerns of myself and the other reviewers.

Reviewer #2 (Remarks to the Author):

My concerns were addressed, and the paper much improved.

Reviewer #3 (Remarks to the Author):

I acknowledge the efforts made by the authors to address my comments. I now understand why the number of strains of isolates was limited, and the authors have justified this. Most minor comments have been satisfactorily addressed but the main concern regarding the conclusions that can be drawn from this small sampling remains. The new data shown by the authors indeed depicts a high variability within pairs of isolates from the same outbreak. Although this variability was significantly smaller within the same outbreak as compared to between outbreaks, it is still large and of a similar size (i.e. 198 vs 245 lost genes, meaning that 80% of the differences between two outbreaks may depend on which isolate from that population they sampled). With such a huge diversity within a single outbreak, and the small number of isolates considered I still think that many of the conclusions presented may be sample-dependent.

Reviewer #1:

R1.1	I am satisfied with the changes made by the authors to address the concerns of myself and the other reviewers.	We thank the reviewer for their time and constructive comments, we think they have much improved the manuscript
------	--	---

Reviewer #2:

R2.1	My concerns were addressed, and the paper much improved.	We thank the reviewer for their time and constructive comments, we also think they have much improved the manuscript
------	--	--

Reviewer #3:

R3.1	I acknowledge the efforts made by the authors to address my comments. I now understand why the number of strains of isolates was limited, and the authors have justified this. Most minor comments have been satisfactorily addressed but the main concern regarding the conclusions that can be drawn from this small sampling remains. The new data shown by the authors indeed depicts a high variability within pairs of isolates from the same outbreak. Although this variability was significantly smaller within the same outbreak as compared to between outbreaks, it is still large and of a similar size (i.e 198 vs 245 lost genes, meaning that 80% of the differences between two outbreaks may depend on which isolate from that population they sampled). With such a huge diversity within a single outbreak, and the small number of isolates	We acknowledge the reviewer's concerns that the greater diversity observed between versus within an outbreak may be an artefact of limited sample size and have added text to this effect: Lines 243-245: "while we see substantial variation within an outbreak site, it appears there is more diversification in the gene arsenal between isolates from different outbreak sites than isolates from the same outbreak site. However, the limited sample size studied here, with only pairs of isolates sequentially obtained from each outbreak site, does limit such comparisons, and it is possible that the apparent greater diversity observed between versus within outbreaks is an artefact of this limited sampling." Lines 293-294: "isolates collected serially from the same outbreak site consistently cluster together, appearing to showing less divergence within- than between- outbreak sites (see S.I Figure 2 and Supplementary Figure 3A). While
------	--	---

considered I still think that many of the conclusions presented may be sample-dependent.

further sampling from within outbreaks would be needed for more definitive conclusions; unfortunately the speed with which *Bsal* decimates populations⁸, with minimal to no recovery years after introduction, heavily impedes sequential sampling.”

- We have toned down the discussion of the clustering of isolates from the same outbreak site:
Lines 82-84: “isolates within “time-series pairs” appear to clustered together, and displaying apparently showed lower variation compared to that observed between outbreak sites (see Supplementary Figures 2 and 10)”
- We have removed the two references to “outbreak-specific” diversity (Lines 26 and Lines 27) from the abstract (there were no other references to outbreak-specific differences in the rest of the manuscript)
- We have changed the title to "Diversity, **multifaceted** evolution, and facultative saprotrophism in a virulent fungal epidemic", removing the reference to the speed of evolution, which was in part calculated based on within outbreak comparisons

We think these statements are supported by the data, where we do consistently see isolates from the same outbreak site clustering together in phylogenetic and principle component analyses, and, despite the small sample size, we consistently observe statistically significantly less variation within outbreaks than between outbreak sites in all measures except pairwise comparison of number of proteins deleted (as highlighted in this comment), which does still show the same pattern with an average 20% more genes deleted in comparisons of isolates from different outbreak sites; but a relatively small amount

		of variation between all isolates as protein coding regions are, as one would expect, more conserved than non-coding regions. We thank the reviewer for their time and constructive comments that we feel have improved the manuscript.
--	--	---